PREPARED FOR SUBMISSION TO JHEP

# Pseudo-periodic map and classification of theories with eight supercharges

**Dan Xie**

*Department of Mathematics, Tsinghua University, Beijing, 100084, China*

ABSTRACT: The classification of local one parameter Coulomb branch solution of theories with eight supercharges is given by assuming that it is given by a genus $g$ fiberation of Riemann surfaces. The crucial point is the fact that certain conjugacy class (so-called pseudo-periodic map of negative type) in mapping class group determines the topological type of the degeneration. The classification of conjugacy class has a simple combinatorial description. Each such conjugacy class gives rise to a dual graph and a 3d mirror quiver gauge theory can be derived, which is then used to identify the low energy theory (assuming generic deformation). Some global Seiberg-Witten geometries are given by using the topological data of the degeneration. The geometric setup unifies 4d $\mathcal{N} = 2$ SCFTs (such as $T_n$ theory and Argyres-Douglas theory), 5d $\mathcal{N} = 1$ SCFTs, 6d $(1, 0)$ SCFTs, 4d IR free theories, and 4d asymptotical free theories in a single combinatorial framework.

## 1   Introduction

One of most important ingredient in solving the Coulomb branch of four dimensional $\mathcal{N} = 2$ theory is the electric-magnetic duality of low energy abelian gauge theory [1], namely, when one go along the special vacua (where there are extra massless particles), the effective photon coupling changes by an element of electric-magnetic duality group [1]. By combining the duality group element around various special vacua (solving a Riemann-Hilbert problem), Seiberg-Witten (SW) successfully solved Coulomb branch of $\mathcal{N} = 2$ $SU(2)$ gauge theory by finding a SW curve [1, 2]: a family of genus one curves $F(x, y, u) = 0$,

---

[1]In the rank one case, this group is $SL(2, Z)$.

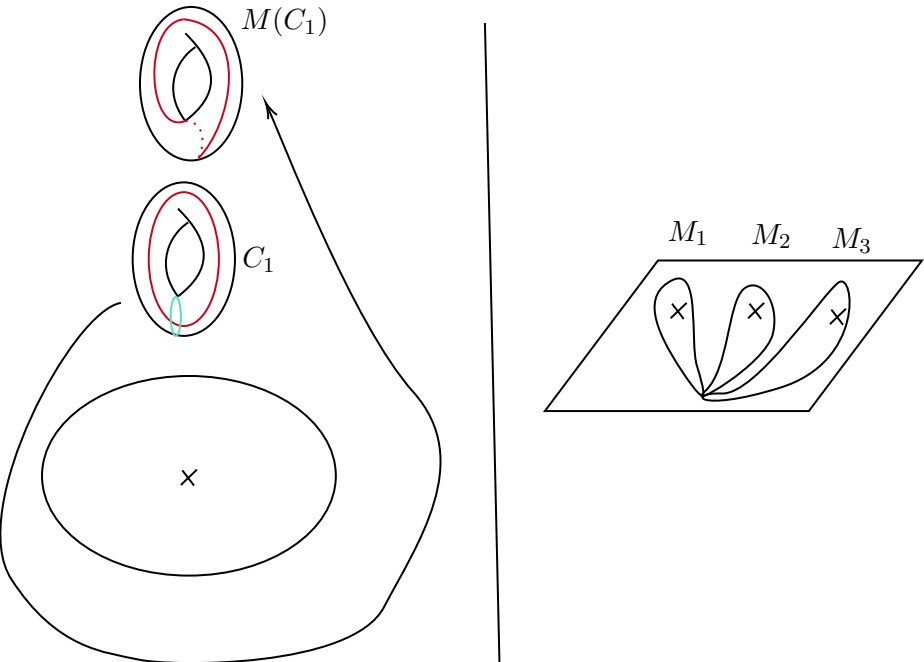

**Figure 1**: Left: The electric-magnetic duality group is given by the monodromy group $M$ which acts on homology group of SW curve; Right: there is one monodromy group element around each special vacua (including the $\infty$ point on the Coulomb branch), and the global constraint is $M_1 M_2 M_3 = I$.

here $u$ is the Coulomb branch parameter. The physical data are extracted as follows: a): The low energy photon coupling $\tau(u)$ is given by the complex structure of smooth curve $F(x, y, u)$; b): The special vacua is given by the degeneration of SW curve; c): The electric-magnetic duality group around it is given by the monodromy group which acts on homology class of the SW curve, see figure. 1.

In practice, one usually solve the Coulomb branch by finding a SW geometry (often using string theory construction), and then try to analyze the IR physics from SW geometry. While the photon couplings at generic vacua can be computed using period integral, the IR physics at special vacua is much more complicated to determine and one usually need to use extra mathematical structure like mixed Hodge structure [3]. One of the findings in [3, 4] is that the electric-magnetic duality (monodromy) group around the special vacua is not enough to find the IR theory. For example, in rank two case [5], besides the monodromy group, one also need to specify two extra set of data ( an integer $m$ and the degeneration type at the singularity) so that the low energy theory can be determined (by assuming generic deformation).

While a large class of SW geometries were already found [6–11], to perform complete classification one need to follow the original approach of SW: namely first classify the special IR vacua and then try to solve the Riemann-Hilbert problem to get the global solution [1, 2]. Certainly, the first and crucial step is to classify the local behavior around

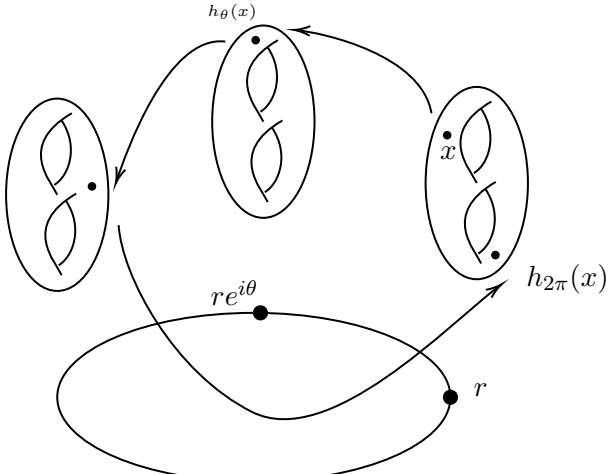

**Figure 2**: There is a homeomorphism action $h$ associated with the loop around special vacua. The action acts on every point on Riemann surface, and in particular it induces an action on homology group.

the special vacua, which have been finished for rank one case [12–14] and rank two case [4].

The purpose of this paper is to classify the local behavior of SW geometry for theory with arbitrary rank $g$. Several assumptions on SW geometry are made: a): The local solution is described by a fiberation of principally polarized abelian varieties [15] which can be identified with Jacobian of genus $g$ Riemann surfaces, and so the local SW solution is given by the fiberation of a Riemann surface and the special vacua is given by the degeneration of Riemann surfaces; b): We take a one parameter slice of Coulomb branch and so there is a one parameter family of Riemann surfaces; c) The theory at special vacua is associated with the generic deformation of the degeneration. Therefore, the classification of local theory is reduced to the problem of classification of one parameter degeneration of Riemann surface.

This task seems quite difficult given the fact that the number of degeneration type increases dramatically with the genus (140 for rank two [5], 1600 for rank three [16]), and it seems to be hard to organize those degenerations even for rank two case. Luckily, Matsumoto-Montesinos [17] (MM) gave a remarkable combinatorial description for such classification. The crucial idea is to use the conjugacy class of **mapping class group** action on the Riemann surface in going around the special vacua, see figure. 2. The mapping class group action certainly determines the action on homology group, and therefore gives the electric-magnetic duality group, but it contains more information, and in particular the degeneration can be completely specified once the conjugacy class in mapping class group is specified [17]!

In the context of degeneration of Riemann surface $\Sigma_g$, the conjugacy class of mapping class group is of special type: it is the so-called pseudo-periodic map of negative type. Those maps are classified by following data:

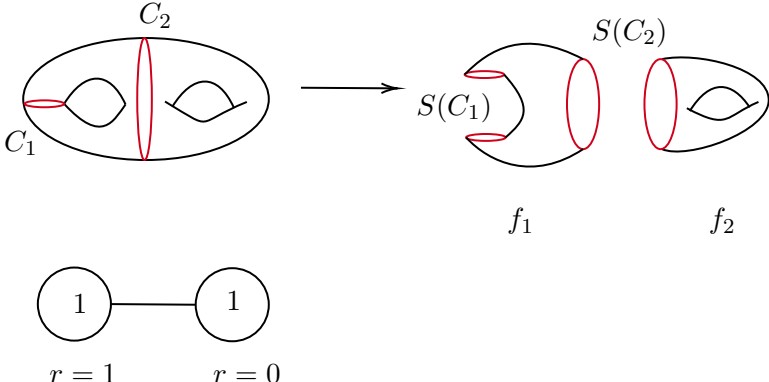

**Figure 3**: Up: An admissible cut system for a genus two Riemann surface, and there are two irreducible components after the cutting; there is a screw number associated with each cutting curve, and a periodic map on each irreducible component. Bottom: A weighted graph for the above cut system: here one draw an edge for every separating curve (which will cut the Riemann surface into two separate components), and one write the number of non-separating cutting curves for a vertex in the graph.

1. An admissible system of cut curves $\mathcal{C} = \cup C_i$ on $\Sigma_g$. Admissible means that the irreducible component of $B = \Sigma_g/\mathcal{C}$ has negative Euler number $\chi_i = 2 - 2g_i - n_i \geq 0$, here $n_i$ is the number of boundary curves for an irreducible component $B_i$ of $B$, and $g_i$ is the genus.

2. The action of $f$ on the oriented graph $G_{\mathcal{C}}$ induced by $\mathcal{C}$.

3. The screw numbers $S(C_i)$ around each annulus $C_i$. Here the screw number is required to be negative.

4. The action $f$ restricted on each irreducible component of $B$ is a periodic map, which is in turn determined by the so-called valency data: $(n, g', \frac{\sigma_1}{\lambda_1} + \frac{\sigma_2}{\lambda_2} + \ldots + \frac{\sigma_s}{\lambda_s})$, here $n$ is the order of the map ($f^n = id$), and $g'$ is the genus of the base defined by the covering map $f : \Sigma \to \Sigma'$, and $\sigma_i, \lambda_i$ are the integral value, see section 2.1.

In summary, one get a weighted graph for first two steps, See figure. 3 for an example. There is also an integer $K \geq -1$ for every annulus (this number is determined by screw number and the boundary data on the annulus); finally, there is a periodic map for each irreducible component in $B$. Therefore, the classification can be done in a combinatorial way!

Once the degeneration is classified, the next question is to find the low energy theory associated with it, which is in general a quite difficult question. It turns out that the dual graph in MM's theory will help us solve this problem. The crucial observation is the link between the dual graph and the 3d mirror of the low energy theory [4]. Notice that one can not always find a 3d mirror from the dual graph, which might give a criteria to determine whether a degeneration can appear in the SW geometry or not. Using the dual graph

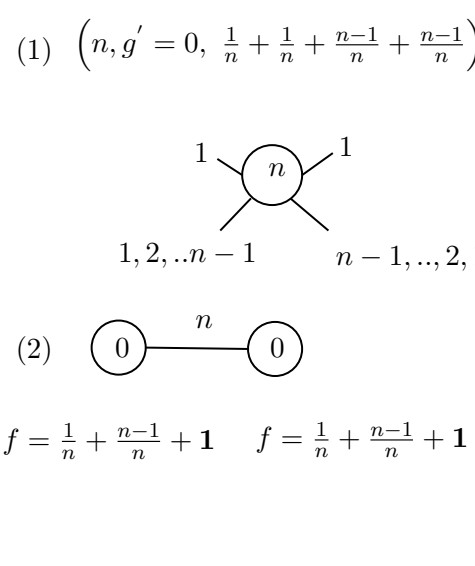

$(1)$ $\left( n, g' = 0, \ \frac{1}{n} + \frac{1}{n} + \frac{n-1}{n} + \frac{n-1}{n} \right)$

$(2)$

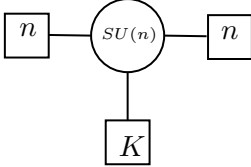

$f = \frac{1}{n} + \frac{n-1}{n} + \mathbf{1} \qquad f = \frac{1}{n} + \frac{n-1}{n} + \mathbf{1}$

**Figure 4**: (1): The data for a periodic map of a genus $n-1$ Riemann surface, and the dual graph for the degeneration is drawn; the dual graph is just the 3d mirror for $SU(n)$ theory coupled with $2n$ fundamental matter [21]; (2): A weighted graph for a genus $n-1$ curve, here $K \geq 1$; the IR theory is $SU(n)$ gauge theory coupled with $2n + K$ hypermultiplets in fundamental representation.

(and the associated 3d mirror) for the degeneration, the low energy theory can be roughly described as follows:

1. One find a four dimensional $\mathcal{N} = 2$ SCFT for periodic map, and many familiar theories such as $(A_{n-1}, A_{k-1})$ theory [7, 18], $T_n$ theory [6], and free bi-fundamental matter can be found. See table. 1 for more examples.

2. The gluing of different components in the weighted graph are interpreted as gauging the flavor symmetry of the matter system.

See figure. 4 for a couple of examples. This type of description of low energy theory is very much like what was used in class $\mathcal{S}$ theory, see figure. 25. In our case, the Argyres-Douglas theory, $T_n$ theory, and IR free gauge theory are described in an unified framework.

Once the local studies are finished, the global SW geometry can be studied by adding a point $\infty$ on the physical Coulomb branch [3]. A singular fiber $F_\infty$ would also be added at $\infty$ which fits into previous local study. The property of $F_\infty$ can be used to determine the UV property. We give simple topological constraints on $F_\infty$ so that it can define the SW geometry of a UV complete 4d, 5d or 6d theory. The results are: a): 4d theory: the dual

graph of $F_\infty$ is a tree of rational curves; b): 5d theory: the dual graph $F_\infty$ is a chain of rational curves with just one loop; c): 6d theory: the dual graph $F_\infty$ is a chain of rational curves with two loops! It is interesting to note the number of loops on the dual graph matches the number of loops in doing the compactification. Some global SW geometries for many familiar theories in 4d, 5d and 6d are discussed, and thorough studies would be left for separate publications.

This paper is organized as follows: section two reviews the pseudo-periodic map; section three discusses how to attach a dual graph for a pseudo-periodic map; section four discusses how to find the low energy theory from the dual graph and related 3d mirror; section five discusses topological constraints on UV singular fiber, and several classes of global SW geometries for 4d, 5d, and 6d theories are given; finally a conclusion is given in section six.

## 2 Pseudo-periodic map

Let's first recall the definition of mapping class group, for more details, see [19]. Let's assume that $S$ is the connect sum of $g \geq 0$ tori with $b \geq 0$ open disks removed and $n \geq 0$ points removed from the interior. Let $Homeo^+(S, \partial S)$ denote the group of orientation-preserving homeomorphisms of $S$ that restrict to the identity of $\partial S$. The mapping class group of $S$, denoted as $Mod(S)$, is the group

$$Mod(S) = Hemeo^+(S, \partial S)/Homeo_0(S, \partial S)$$

Here $Homeo_0(S, \partial S)$ denotes the connected component of the identity in $Hemeo^+(S, \partial S)$. For example, the mapping class group of a closed genus one Riemann surface is $SL(2, Z)$. There are also other equivalent definition of mapping class group, i.e. the isotopy class of $Hemeo^+(S, \partial S)$.

We also need to consider the generalization of above mapping class group, namely, the homeomorphism $f$ might just keep the boundary $\partial S$ as a set, and more generally, $f$ could also permute the elements in $\partial S$.

As discussed in the introduction, a local SW solution of the particular type [2] gives rise to a one parameter family of genus $g$ fiberation. If there is a special vacua, then one can associate an element of mapping class group, see figure. 2. Actually, one associate a conjugacy class by changing the base point of the loop around the singularity [17, 20].

Instead of starting with known SW solution and try to compute the mapping class group element around the degeneration, we go in the opposite direction. We first try to classify all possible topological type of one parameter degeneration of genus $g$ fiberation [3], and then try to determine the IR theory for each degeneration.

The classification of one parameter degeneration of Riemann surface has been found in [17, 20], whose main results are: First, the conjugacy class realized as the monodromy of a degeneration of curves (with genus $g \geq 2$) is represented by a pseudo-periodic map of **negative** type; Secondly, any conjugacy class of pseudo-periodic map of negative type is realized as the topological monodromy of a certain degeneration of curves.

Therefore the classification of the degeneration is reduced to that of the pseudo-periodic map $f$ of negative type, which has a combinatorial description.

1. First there is an admissible system of cut curves $\mathcal{C} = \cup C_i$. Here admissible means that each irreducible component of $B = \Sigma_g/\mathcal{C}$ has negative Euler number $\chi_i = 2 - 2g_i - n_i < 0$, here $n_i$ is the number of boundary curves for an irreducible component $B_i$ of $B$. One can associate a graph consists of vertices and edges: the vertex has label of genus and the non-separating cut curves, and the edges denote the separating curves. This graph is denoted as $G_\mathcal{C}$.

2. There is a finite (cyclic) group action $\sigma$ on the oriented graph $G_\mathcal{C}$ induced by the map $f$. One can get a weighted graph $Y$ by adding weights to vertices and edges of the quotient graph of $G_\mathcal{C}$ by $\sigma$.

---

[2] If the associated abelian variety is given by the Jacobian of a Riemann surface.

[3] Let's emphasize that unlike rank one case, it appears that some degenerations can not appear in the Coulomb branch solution.

3. The screw numbers $S(C_i)$ of $f$ around each annulus $C_i$ in the cut system. Here the screw number $S(C_i)$ is required to be negative.

4. The action $f$ restricted on each component of $B$ is a periodic map, which is in turn determined by the so-called valency data.

Let's now explain above ingredients in more details:

*Cut system of curves*: The classification of an admissible system of cut curves $\mathcal{C}$ is the same as classification of the stable curves introduced by Deligne-Mumford. Such cut systems can be represented by a weighted graph as follows: a) The vertex $v$ represents an irreducible component, and two integers $(g(v), \rho(v))$: $g(v)$ the genus of $v$, $\rho(v)$ the number of cut curves which only belongs to $v$. b) The edge $e$ represents the separating cut curves, see figure. 5 for an example.

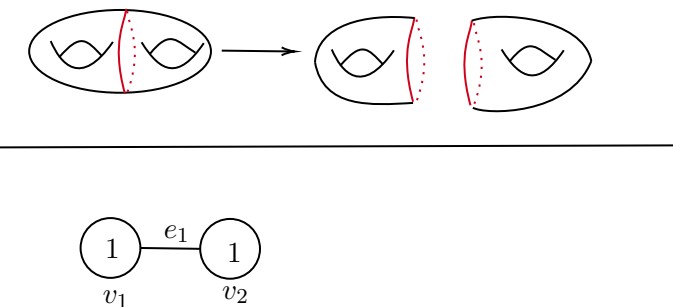

**Figure 5**: Up: There is only one cut curve for a genus two Riemann surface; There are two components after the cut, and each component has genus one and zero internal (non-separating) cut curves; Bottom: The graph for the cut system.

*Automorphism*: Given a cut system and so a graph $X$. By an automorphism $\sigma : X \rightarrow X$, we mean an automorphism of the graph such that the weight $(g(B_v), \rho(v))$ is the same as $(g(B_{\sigma(v)}), \rho(\sigma(v)))$ for each vertex $v$ of $X$. For example, for $X$ listed in figure. 5, there is a $\mathbb{Z}_2$ action exchanging $v_1$ and $v_2$, and fixing the edge $e_1$. We can then get a weighted graph $Y$ by recording the dimension of the orbits of $\sigma$, see section. 2.2 for more details.

*Screw number*: For a curve $\vec{C}_i$ of the cut system $\mathcal{C}$, there exists a minimal integer $\alpha_i$ such that $f^{\alpha_i}(\vec{C}_i) = \vec{C}_i$ (they are equal as a set). There also exists a minimal integer $L_i$ such that $f^{L_i}|_{C_i}$ is a Dehn twist of $e_i$ times ($e_i$ could be positive or negative integers). The screw number is defined as

$$s(C_i) = e_i \alpha_i / L_i.$$

$f$ is called **negative** type if $S(C_i) < 0$ for all the cut curves.

The cut curves are classified into two types. A curve $C_i$ is called **amphidrome** if $\alpha_i$ is even and $f^{\alpha_i/2}(\vec{C}_i) = -\vec{C}_i$, which means the orientation is changed. Otherwise it is **non-amphidrome**.

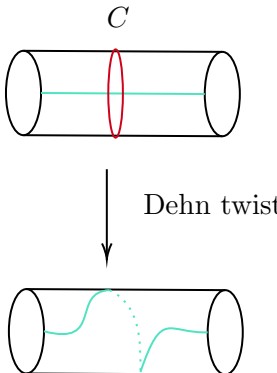

$C$

Dehn twist

**Figure 6**: Dehn twist along curve $C$, and this action is taken to be negative around the indicated direction.

*Valency data*: The valency data for periodic map is more complicated to define. Let $\Sigma$ be a Riemann surface, and $f : \Sigma \to \Sigma$ be a periodic map (namely, $f^n$ is an identity map). For any point $P$ on $\Sigma$, there exists an integer $\alpha(P)$ such that $P, f(P), \ldots, f^{\alpha(P)-1}(P)$ are distinct points, and $f^{\alpha(P)}(P) = P$. If $\alpha(P) = n$, $P$ is called **simple** points, and P is called **multiple** point if $\alpha(P) < n$.

Let $\vec{C}$ be an oriented simple closed curve, and let $m$ be the smallest positive integer such that $f^m(\vec{C}) = \vec{C}$ as a set, and $n = m\lambda$. The restriction of $f$ on a $\vec{C}$ is then a periodic map. Let $Q$ be any point on $C$, the orbit of $Q$ under the iteration of the action $f^m$, and the points are **ordered** as $(Q, f^{m\sigma}(Q), f^{2m\sigma}(Q), \ldots, f^{(\lambda-1)m\sigma}(Q))$, here $\sigma$ is an integer which satisfies the condition $0 \leq \sigma \leq \lambda - 1$, and $gcd(\sigma, \lambda) = 1$. Finally, one introduce another integer $\delta$ which satisfies the condition

$$\sigma\delta = 1(mod\lambda), \quad 0 \leq \delta \leq \lambda - 1.$$

This means that the action $f^m$ on $\vec{C}$ is rotation with angle $2\pi\frac{\delta}{\lambda}$. The data $(m, \lambda, \sigma)$ is the valency data associated with a curve $\vec{C}$. The valency data for a multiple point $P$ is defined as that of the **closed curve around** $P$.

Given a periodic map $f$, one has a $n$-fold cyclic covering map $\Pi : \Sigma \to \Sigma'$ associated with $f$, and $\Sigma'$ is the quotient space with respect to $f$ (the space of orbits). The multiple points of $f$ gives rise to ramification points of $\Pi$. The valency data of the ramification point is defined as that of the multiple points. So finally, a periodic map is specified by the genus of $\Sigma'$ and the valency data for the ramification points.

**Induced action on Homology group**: One of the fundamental aspects of $Mod(S_g)$ is its action on $H_1(S_g, Z)$, and the representation $\Psi : Mod(S_g) \to Aut(H_1(S_g, Z))$ is the first approximation to $Mod(S_g)$. Any $\phi \in Homeo^+(S_g)$ induces an automorphism $\phi_* : H_1(S_g, Z) \to H_1(S_g, Z)$, and it follows that the map

$$\Psi_0 : Mod(S_g) \to Aut(H_1(S_g; Z))$$

has the property that

$$\Psi_0(Mod(S_g)) \subset Sp(2g, Z)$$

Let's now describe the action of Dehn twist on the homology class. Let $a$ and $b$ be isotopy classes of oriented simple closed curves in $S_g$. For any $k \geq 0$, we have

$$\Psi_0(T_b^k)([a]) = [a] + k\hat{i}(a,b)[b].$$

Here $\hat{i}(a,b)$ are the intersection number of $a, b$. Using the above formula, one can see that the Dehn twist acts trivially on separating curves, as the homology class of those curves is trivial. Those mapping class elements which act trivially on homology groups are called Torreli groups.

## 2.1 Periodic map

The periodic map of Riemann surface plays a fundamental role in the classification of the degeneration of the Riemann surface, as they specify the data on the irreducible component after the cut. Let's review the basic facts about these periodic maps.

Let $\Sigma_g$ be a closed Riemann surface of genus $g \geq 2$, and $f : \Sigma_g \to \Sigma_g$ be a cyclic analytic automorphism of order $n$, and $\Pi : \Sigma \to \Sigma'$ be the $n$-fold cyclic covering. Let $g'$ be the genus of $\Sigma'$, and $\lambda_1, \ldots, \lambda_l$ be the ramification indices of $\Pi$ and $(n/\lambda_i, \lambda_i, \sigma_i)$ be the valency data. The valency data has to satisfy following conditions:

1. The Hurwitz formula: this formula relates the genus $g$ to the valency data of ramification points:

$$2g - 2 = n[2g' - 2 + \sum_{i=1}^{l}(1 - \frac{1}{\lambda_i})]. \tag{2.1}$$

2. Nielson theorem: $\sum \sigma_i/\lambda_i$ is an integer. This means that the formal sum of valency data should be integer, and the minimal number is one.

3. Wiman: This constraint means that the order of the periodic map is constrained by the genus: $n \leq 4g + 2$.

4. Havey: This constraint is more complicated to stay, and it put the constraint on the number of ramification points. Set $M = lcm(\lambda_1, \ldots, \lambda_l)$ [4].

   (a) $lcm(\lambda_1, \ldots, \hat{\lambda}_i, \ldots, \lambda_l) = M$ for all $i$, here $\hat{\lambda}_i$ denotes the omission of $\lambda_i$. This constraint implies that the lcm of denominator of the valency data should be the same by omitting just one ramification point.

   (b) $M$ divides $n$, and if $g' = 0$, then $M = n$.

   (c) The number of ramification point is greater than one: $l \neq 1$. Moreover, for $g' = 0$, there should be at least three ramification points.

   (d) If $M$ is an even number and so $M = 2^\delta M_1$ with $M_1$ an odd number, then the number of $\lambda_i$ which are divisible by $2^\delta$ is even.

---

[4] Here $lcm$ denotes the least common multiple of the numbers within bracket.

Usually, one label the periodic map by the data $(n, g', \frac{\sigma_1}{\lambda_1} + \frac{\sigma_2}{\lambda_2} + \ldots + \frac{\sigma_l}{\lambda_l})$. On the other hand, given a data satisfying above condition, one can find a period map of the Riemann surface $\Sigma_g$.

*Example 1*: The periodic map is given by $(n, g' = 0, \underbrace{\frac{1}{n}, \ldots, \frac{1}{n}}_{a}, \frac{n-a}{n})$, $a \geq 2$. If the common divisor of $n$ and $a$ is $b$, the genus of $\Sigma_g$ (see formula. 2.1) is

$$2g - 2 = n[-2 + \sum_{i=1}^{a}(1 - \frac{1}{n}) + (1 - \frac{1}{n/b}] =$$
$$(a-1)n - a - b \rightarrow g = \frac{(a-1)n - a - b + 2}{2}.$$

In particular, if $b = (n, a) = 1$, the genus $g = \frac{(a-1)(n-1)}{2}$.

*Example 2*: The periodic map is given by $(n, g' = 0, \underbrace{\frac{n-1}{n}, \ldots, \frac{n-1}{n}}_{n})$. The genus of $\Sigma_g$ is

$$2g - 2 = n(n - 3) \rightarrow g = \frac{(n-1)(n-2)}{2}.$$

**Duality for periodic map**: Given a period map, one can change the valency data $\frac{\sigma_i}{\lambda_i} \rightarrow \frac{\sigma_i^d}{\lambda_i} = \frac{\lambda_i - \sigma_i}{\lambda_i}$ to define another periodic map. So the following two periodic maps are dual to each other:

$$(n, g', \frac{\sigma_1}{\lambda_1} + \frac{\sigma_2}{\lambda_2} + \ldots + \frac{\sigma_l}{\lambda_l}) \rightarrow (n, g', \frac{\lambda_1 - \sigma_1}{\lambda_1} + \ldots + \frac{\lambda_l - \sigma_l}{\lambda_l}).$$

**Periodic map for surface with boundaries**: For a genus $g$ surface with $k$ boundaries, the period map is defined as the one associated with closed genus $g$ surface by adding disks around $k$ boundaries. One often need to specify the transformation properties of the boundary curves. Here is an example, let's take the order of $f$ to be 4, and $f$ permute the boundaries as $(\partial_1, \ldots, \partial_4)$ (so $m = 4$ for these points), and so under the coving map, the center of these boundaries are mapped to a single point on $\Sigma'$, and the associated valency data is $(m, \lambda, \sigma) = (4, 1, 1)$, with $\frac{\sigma}{\lambda} = 1$.

## 2.2 Weighted graph

Given a cut system $X$ of Riemann surface $\Sigma_g$ and an automorphism $\sigma$, one can define a weighted graph $Y$ from $X$ as follows. We need to enlarge the action by adding an orientation for each edge in $X$ (one can just conveniently add arrows for the edges). Then one have more automorphism by requiring it to perserve the orientations.

The weighted graph $Y$ is derived from $X$ and $\sigma$ as follows. Let $h : X \rightarrow Y$ be the quotient map defined by $\sigma$, and $Y$ is constructed as follows: a): let $\bar{v}$ be a vertex of $Y$, then $h^{-1}(\bar{v})$ consists of $l$ vertices (their weights $(g(v_i), r(v_i))$ coincide with each other) of $X$ which are permuted by $\sigma$; so vertex $\bar{v}$ has the triple weight $(l(\bar{v}), g(\bar{v}), r(\bar{v}))$; b) let $\bar{e}$ be an edge of $Y$; Then $h^{-1}(\bar{e})$ consists of a finite number $\zeta(\bar{e})$, of edges of $X$, so there is a

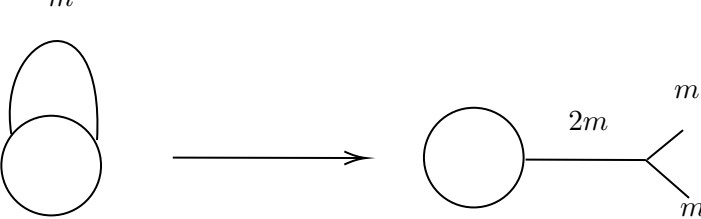

**Figure 7**: The loop in the weighted graph $Y$ is resolved to a D type graph.

weight $\zeta(\bar{e})$ for $\bar{e}$. Notice that if there is an edge $e$ in $X$ connecting $v$ and $v^{'}$, and such that $h(v) = h(v^{'})$, then there would be a loop in $Y$.

Finally one need to define the resolution graph $\tilde{Y}$ of $Y$ by modifying the loops in $Y$. This happens in the following situation: Suppose that there is an edge of $X$ and a positive integer $m$ such that

$$\sigma^m(e) = e, \sigma^m(v) = v^{'}, \sigma^m(v^{'}) = v.$$

Here $e$ connects $v$ and $v^{'}$. Notice that this condition does not include the orientation of the edges, namely $\sigma^m(e) = e$ without considering the orientation. The above condition means that $\sigma^m$ fixes $e$ and exchanges the vertex $v$ and $v^{'}$. Let $m$ be smallest integer which satisfies above condition, then one replace the loop in $Y$ by a D type graph with weights $(2m, m, m)$, see figure. 7.

*Example*: Let's consider a cut system for genus two Riemann surface, and it is given by a graph with two vertex with genus 0 and three edges between them, see the bottom of figure. 8, and here we choose orientations for the edges in the cut system. There are following cyclic automorphism [5] for it:

1. $\sigma$ acts trivially.

2. $\sigma$ fixes $v_1$ and $v_2$, and exchanges $e_1$ and $e_2$.

3. $\sigma$ fixes $v_1$ and $v_2$, and cyclic permute $e_1, e_2, e_3$.

4. $\sigma$ exchanges $v_1$ and $v_2$, fixes $e_1, e_2, e_3$ as sets, but with their orientations reversed. We find that $\sigma(e_i) = e_i$ (as a set), and $\sigma(v_1) = v_2, \sigma(v_2) = v_1$, and so one need to do resolutions on the loops in the weighted graph.

5. $\sigma$ exchanges $v_1$ and $v_2$, and exchanges $e_1$ to $-e_2$, and $e_2$ to $-e_1$, $e_3$ to $-e_3$. Here $-e_i$ means $e_i$ with orientation reversed.

6. $\sigma$ exchanges $v_1$ and $v_2$, and exchanges $e_1$ to $-e_2$, and $e_2$ to $-e_3$, $e_3$ to $-e_2$.

So there are a total of six types of weighted graph, see the bottom of figure. 8. The interested reader might try to match the weighted graph with above automorphism action.

---

[5]By automorphism, we mean the resulting graph after the specified action is identified with the original graph by renaming the edges.

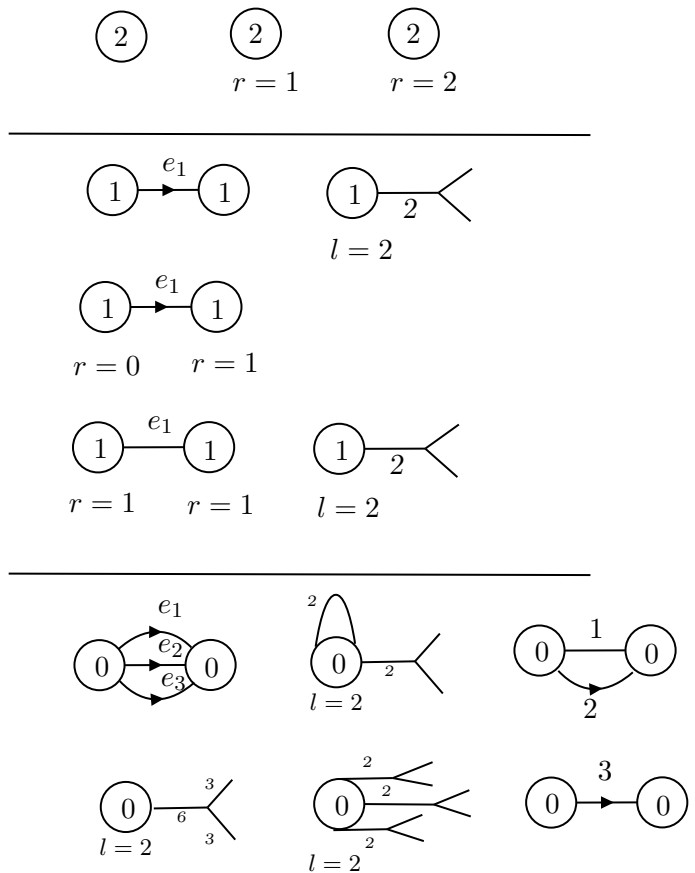

**Figure 8**: All possible weighted graph for genus two Riemann surface.

## 3 Dual graph

The pseudo-periodic map of negative type have been described in last section. One can attach a dual graph for each conjugacy class which will play a crucial role in our later study of physical theories.

### 3.1 Dual graph for period map

First, one can define a star-shaped dual graph for a period map as follows. A periodic map is given by the data $(n, g', \frac{\sigma_1}{\lambda_1} + \ldots + \frac{\sigma_l}{\lambda_l})$, and the dual graph is constructed as follows:

1. First given the valency data $\frac{\sigma}{\lambda}$ $(m\lambda = n)$, one attach a linear chain of spheres with following nonzero multiplicities $a_0 > a_1 > a_2 \ldots > a_s = 1$:

$$a_0 = \lambda, \quad a_1 = \sigma, \quad \frac{a_{i+1} + a_{i-1}}{a_i} = \lambda_i \in Z.$$

Given $a_i$ and $a_{i-1}$, the above formula uniquely determines the integer $a_{i+1}$ by assuming $\lambda_i$ to be the minimal integer satisfying the equation. Since $n = \lambda m$, the final chain of spheres for the valency data $\frac{\sigma}{\lambda}$ are

$$\boxed{ma_0 - ma_1 - ma_2 - \ldots - ma_{s-1} - m.}$$

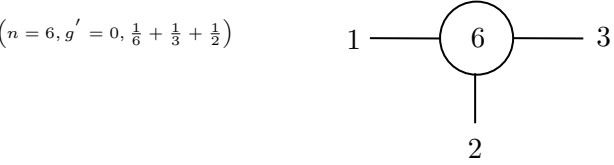  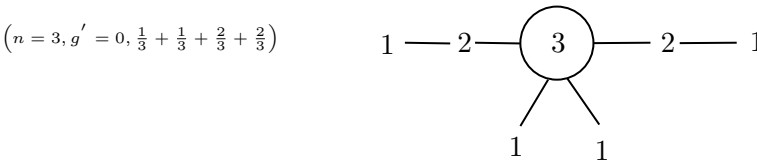

**Figure 9**: The dual graph for periodic maps.

*Example 1*: If the data is $n = 7, \frac{\sigma}{\lambda} = \frac{5}{7}$, the tail is $7 - 5 - 3 - 1$; On the other hand, for the data $n = 6, \frac{\sigma}{\lambda} = \frac{2}{3}$, the tail is $6 - 4 - 2$.

*Example 2*: If the data is $\frac{n-1}{n}$. The tail would be

$$n - (n - 1) - (n - 2) - \ldots - (2) - 1$$

Notice that this is the quiver tail corresponding to the so-called full regular puncture of $A_{n-1}$ type [21], here we regard the multiplicity as the rank of the gauge group of the corresponding quiver gauge theory. This identification will play an important role later.

*Example 3*: If the data is $\frac{1}{n}$. The tail would be

$$n - 1$$

This is the quiver tail corresponding to the so-called simple regular puncture of $A_{n-1}$ type [21].

2. The final graph is formed as follows: one has a core component with genus $g'$, and the multiplicities $n$, then attach the linear tail from each ramification point. The gluing is simple as the first component of each tail has the common multiplicities $n$. At the end, one get a star-shaped graph!

   *Example 1*: The data for the periodic map is $(n = 6, g' = 0, \frac{1}{6} + \frac{1}{3} + \frac{1}{2})$, and the dual graph is shown in figure. 9.

   *Example 2*: The data for the periodic map is $(n = 3, g' = 0, \frac{1}{3} + \frac{1}{3} + \frac{1}{3} + \frac{2}{3})$, and the graph is shown in figure. 9. Notice that the dual graph is the 3d mirror for $SU(3)$ gauge theory with $n_f = 6$.

## 3.2 Gluing

Let's now construct dual graph for general pseudo-periodic map. The first data is a weighted graph $Y$, and the irreducible components in $Y$ are given by genus $g$ Riemann surface with $k$ boundary curves (Let's use $\partial_i$ to denote them.), and $r$ internal non-separating

cutting curves (Let's use $C_i$ to denote them.). There is an associated periodic map on each irreducible component. More explanation on the action of $f$ on the boundary curves is needed (Here $n$ is the order of $f$):

1. First, one need to specify the action of $f$ on boundary curves $\partial_i$, for example $f$ could permute $(\partial_1, \partial_2, \partial_3)$, this implies that branch points of $\Sigma'$ associated with three boundary curves has the data $m = 3$, and so $\lambda = \frac{n}{3}$ for the corresponding valency data.

2. Secondly, one need to specify the action of $f$ on the internal cutting curve $C_i$. There are two types of action according to whether the action is amphidrome or not. Let's take two curves as an example: a): if $C_1, C_2$ is non-amphidrome, and the boundary curves are $C_1', C_1'', C_2', C_2''$, then $f$ permutes $C_1', C_2'$ and so the branch point $P_1$ corresponding to them has $m = 2$, similarly $f$ also permute $C_1'', C_2''$ (with $m = 2$), and branch point is $P_2$ with $m = 2$. One then specify the valency data for $P_1$, $P_2$, and in particular $\lambda = \frac{n}{2}$. In general, if $\langle C_1, \ldots, C_s \rangle$ is permuted by $f$, then there are **two** branch points on $\Sigma'$, and the order of them is $m = s$.

   If the action on $C_1$ is amphidrome, then $f$ permutes two boundary curves $C_1', C_1''$ and so $m = 2$. Similarly, if the action on $C_1, C_2$ is amphidrome, namely, $f(\vec{C}_1) = -\vec{C}_2, f^2(\vec{C}_1) = -\vec{C}_1$, so the boundary curves $(C_1', C_1'', C_2', C_2'')$ is permuted by $f$, then $m = 4$ for these boundary curves. In general, if $\langle C_1, \ldots, C_s \rangle$ is amphidrome under $f$, then there is just **one** branch point on $\Sigma'$, with order $m = 2s$.

**Remark**: We will use bold number for valency data for the boundary curves, and use bracket for the valency data of the internal cutting curves (two for non-amphidrome action, and one for amphidrome action). For exmaple, if $g = 0, k = 4, r = 0$, we have following possibilities: a): $f = id$; b): $\langle \partial_1, \partial_2 \rangle, f = \mathbf{1} + \frac{1}{2} + \frac{1}{2}$; c): $\langle \partial_1, \partial_2 \rangle, \langle \partial_3, \partial_4 \rangle, f = \mathbf{1} + \mathbf{1} + \frac{1}{2} + \frac{1}{2}$; d): $\langle \partial_1, \partial_2, \partial_3 \rangle, f = \mathbf{1} + \frac{1}{3} + \frac{2}{3}$ or $\mathbf{1} + \frac{2}{3} + \frac{1}{3}$; e): $\langle \partial_1, \partial_2, \partial_3, \partial_4 \rangle, f = \mathbf{1} + \frac{1}{4} + \frac{3}{4}$.

In last subsection, one associate a dual graph for each periodic map. Now one can get a dual graph for any pseudo-periodic map by gluing the graphs for period maps together. Let $A_i$ be an annular neighborhood of $C_i$ (possibly many curves, see the discussion above). Let's also use $(m^{(1)}, \lambda^{(1)}, \sigma^{(1)})$ and $(m^{(2)}, \lambda^{(2)}, \sigma^{(2)})$ to be the valencies of the boundary curves $C_i'$ and $C_i''$.

Let's first assume $C_i$ to be non-amphidrome. Then $m^{(1)} = m^{(2)} = m$ so that one can glue them together, and one obtain two sequences of integers $a_0 > a_1 > \ldots > a_u = 1$ and $b_0 > b_1 > \ldots > b_v = 1$. Graphically, one get two quiver tails from above sequence of integers. Define an integer

$$K = -s(C_i) - \delta^{(1)}/\lambda^{(1)} - \delta^{(2)}/\lambda^{(2)} \tag{3.1}$$

where $\delta^{(j)}$ are integers such that $\sigma^{(j)}\delta^{(j)} = 1 (mod \lambda^{(j)})$, $0 \leq \delta^{(j)} < \lambda^{(j)} - 1$. If $\lambda^{(j)} = 1$, one set $\delta^{(j)} = 0$. $K$ satisfies condition $K \geq -1$, as $s(C_i) < 0, 0 \leq \delta^{(j)}/\lambda^{(j)} < 1$. The gluing for the two quiver tails is defined as follows

1. If $K \geq 1$, the the glued tail looks as follows

$$(ma_0, ma_1, \ldots, ma_u, \underbrace{m, m, \ldots, m}_{K-1}, mb_v, \ldots, mb_1, mb_0)$$

2. If $K = 0$, the the glued tail looks as follows

$$(ma_0, ma_1, \ldots, ma_{u-1}, m, mb_{v-1}, \ldots, mb_1, mb_0)$$

3. Finally, if $K = -1$, then one can find $u_0 < u$ and $v_0 < v$ so that $a_{u_0} = b_{v_0}$, and $(a_{u_0-1} + b_{v_0-1})/a_{u_0}$ is an integer greater than one. Then the quiver tail looks like

$$(ma_0, ma_1, \ldots, ma_{u_0}, mb_{v_0-1}, \ldots, mb_1, mb_0)$$

Let's now assume $C_i$ to be amphidrome, then $C_i', C_i''$ has valency data $(2m, \lambda, \sigma)$. Similarly, one has a sequence of integers $a_0 > a_1 > \ldots > a_u = 1$, from which one can get a quiver tail. Then $K = -s(C_i)/2 - \delta/\lambda$ is a non-negative integer where $\delta\sigma = 1(mod\lambda)$. The glued quiver tail now has $u + K + 2$ spheres, and it is a Dynkin diagram of $D$ type

$$(2ma_0, 2ma_1, \ldots, 2ma_u, \underbrace{2m, \ldots, 2m}_{K} \text{(the tree part)}, \ m, m \text{ (the terminal part)})$$

*Example1*: Let's take $C_i$ to be non-amphidrome, and the valency data for both of them is $(m, \lambda, \sigma) = (1, 3, 2)$, so $\delta^1 = \delta^2 = 2$; the integer $K = -S(C_i) - \frac{2}{3} - \frac{2}{3} \geq -1$. The glued tail is shown in figure. 10.

*Example2*: Let's take $C_i$ to be amphidrome, and the valency data is $(m, \lambda, \sigma) = (2m, n, n-1)$ ($\frac{\sigma}{\lambda} = \frac{n-1}{n}$, $m = 1$), and the integer $K = -S(C_i) - \frac{n-1}{n} \geq 0$. The glued tail is shown in figure. 10.

### 3.3 Dual graph for arbitrary pseudo-periodic map

As discussed in last section, the conjugacy class of pseudo-periodic map is classified by weighted graphs. There is an associated periodic map for each vertex $v$ in the graph $\tilde{Y}$, and $e_1, \ldots, e_s$ are the edges on $v$, $e_{s+1}, \ldots e_{s+s'}$ are the loops. So a vertex of $\tilde{Y}$ has data $(l(v), g(v), \rho(v))$ and the edges $1 \leq i \leq s + s'$ with weight $\zeta(e_i)$, and the edge $e_i''$ ($s + 1 \leq i \leq s + s'$) has weights $\zeta(e_i)$. One associate a curve with following data:

$$g = g(v), \quad r = \rho(v), \quad k = \frac{\sum_{i=1}^{s+s'} \zeta(e_i') + \sum_{i=s+1}^{s+s'} \zeta(e_i'')}{l(v)}$$

Namely there are a total of $s + 2s'$ tails. Write the dual graph for the component (with data $(g(v), \rho(v), k(v))$ as follows:

$$S_f = \sum_j m_j E_j + \sum_i n_i \vec{F_i}$$

Then the dual graph is found by replacing the component $\sum m_j E_j$ by $\sum l(v) m_j E_j$. Essentially, one need to multiply $l$ for every component in the dual graph of the vertex $(g, r, k)$.

**Example**: Consider weighted graph shown in figure. 12, and the data for associated curve has $g = 1, r = 0, k = 2$, and $f$ keeps $\partial_1, \partial_2$ separately (so $m = 1$ for them). The data for periodic map is taken to be $\frac{3}{4} + \frac{3}{4} + \frac{1}{2}$. See the dual graph in figure. 12.

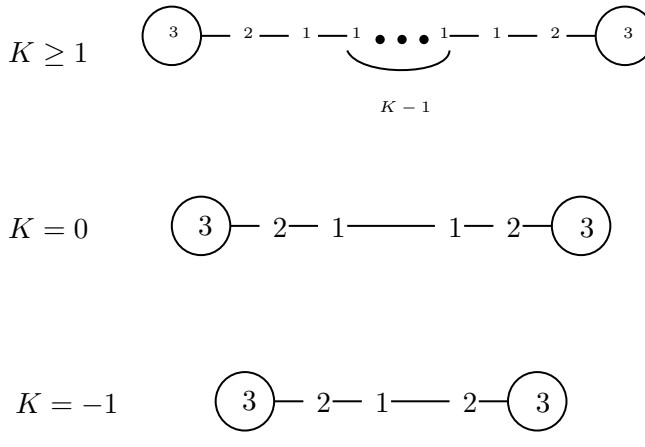

**Figure 10**: Up: The glued tail for a cylinder with same valency data $(m, \lambda, \sigma) = (1, 3, 2)$, and $K = -S(C) - \frac{2}{3} - \frac{2}{3} \geq -1$. Bottom: The glued tail for the amphidrome cut, here the valency data is $(m, \lambda, \sigma) = (2, n, n - 1)$.

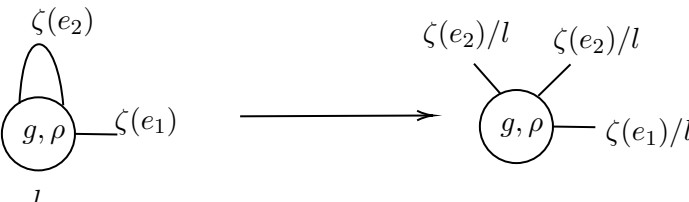

**Figure 11**: For a component in weighted graph, one get an associated genus $g$ curve with $k$ boundaries, and the multiplicity of the boundary edges is divided by $l$.

## 4   IR theory

Let's determine the 4d IR theory associated with a conjugacy class of pseudo-periodic map. Specifying only the conjugacy class is not enough to determine the IR theory, as there would be more than one theory associated with a single degeneration, see [12] for rank one example. The ambiguity can be fixed if one further assume the IR theory is associated with the generic deformation of the singularity. To identify the IR theory, one need to use the interesting relation between the 3d mirror of the IR theory and the dual graph studied in last section [4]. By using the dual graph and the related 3d mirror [7, 21–23], one can

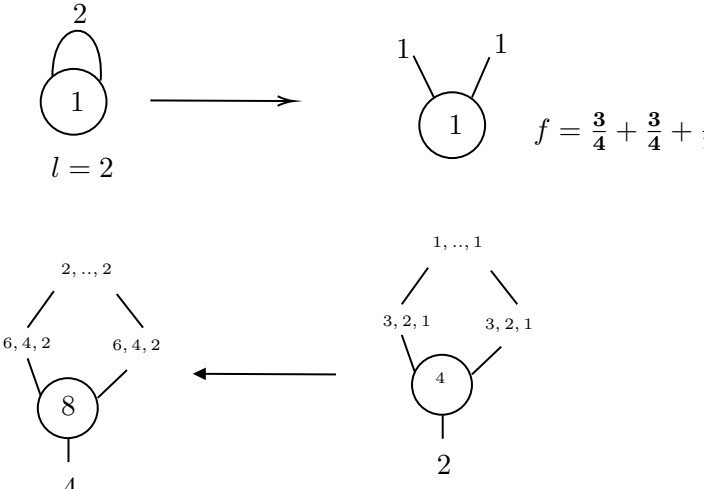

**Figure 12**: Up: A weighted graph with $l = 2$ and a loop with weight 2. The associated curve with boundaries are shown on the right, and the associated periodic map is given too; Bottom: We first write the dual graph for the periodic map, and then get the dual graph of original theory by multiplying two on the multiplicity of all the nodes.

determine the IR theory effectively.

### 4.1 Dual graph and 3d mirror

Let's explain how to define a quiver gauge theory from the dual graph corresponding to a pseudo-periodic map. Given a conjugacy class of pseudo-periodic map, one can get a dual graph:

$$F = \sum n_i C_i;$$

Here $C_i$ is an irreducible curve and $n_i$ is the multiplicity. The intersection number $C_i \cdot C_j$ and the genus $g_i$ are also specified. The self-intersection number for a component $C_i$ can be computed by requiring:

$$C_i \cdot F = 0.$$

The intersection number $k_i = C_i \cdot K$ [6] can be computed from the genus $g_i$ and self-intersection number $C_i^2$:

$$1 + \frac{1}{2}(C_i^2 + k_i) = g_i \rightarrow k_i = 2(g_i - 1) - C_i^2. \tag{4.1}$$

The genus of the configuration $F$ is expressed by the data $k_i$ as follows:

$$1 + \frac{1}{2} \sum n_i k_i = g.$$

See [24] for more details on configuration of curves.

---

[6] Here $K$ is the canonical class of the surface, and only play a formal role here.

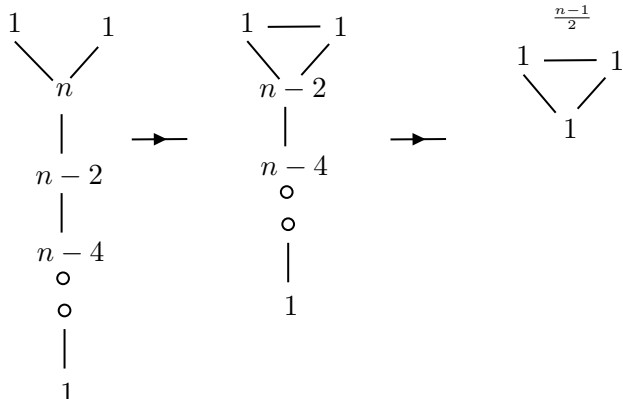

**Figure 13**: Left: the dual graph for a periodic map with data $(n, g' = 0, \frac{1}{n} + \frac{1}{n} + \frac{n-2}{n})$, and node $n$ is a -1 curve. Middle: Contracting node $n$, and there would be one edge between the three nodes connecting node $n$; After contraction, the node with multiplicity $n - 1$ becomes a $-1$ curve; right: Continuing the contracting process, one finally get a model without any $-1$ curve.

**Contracting $-1$ curve**: The dual graph studied in last section is called normally minimal model, since the intersections of components are all double points. There could be $-1$ rational curves (self-intersection number is $-1$) in these models. To get information for the low energy theory, one need to contract $-1$ curve to get a so-called relatively minimal model (no $-1$ curve in the model). The contraction formula is given as follows. Assuming that a $-1$ curve $C_n$ is contracted, and the new intersection number and genus are given as

$$C'_i \cdot C'_j = C_i \cdot C_j + (C_i \cdot C_n)(C_j \cdot C_n),$$
$$g(C'_i) = g(C_i) + \frac{1}{2}((C_i \cdot C_n)^2 - C_i \cdot C_n). \tag{4.2}$$

*Example*: Let's consider a periodic map with data $(n, g' = 0, \frac{1}{n} + \frac{1}{n} + \frac{n-2}{n})$, and $n$ is odd. The dual graph is shown in figure. 13, and all the components are rational curves. The self-intersection number of node $n$ is computed as:

$$C_i \cdot F = 0 \to C_i \cdot (nC_i + \delta) = 0 \to nC_i^2 + C_i \cdot \delta = 0 \to C_i^2 = -1.$$

Here $\delta = F - nC_i$ (which are the components of $F$ minus the central node), and $C_i \cdot \delta = n$ is used. So the central node is a $-1$ curve, which can be contracted and one get a triangle connecting nodes with multiplicity $1, 1, n-2$. After contraction, the node with multiplicity $n-2$ becomes a $-1$ curve; One continue doing the contraction until getting a model without any $-1$ curve, see figure. 13. For our later purpose, we draw the contracting process using the curves, see figure. 14.

**3d Mirror from relative minimal model**: Once the relative minimal model $F = \sum n_i C_i$ is found, a 3d $\mathcal{N} = 4$ quiver gauge theory can be found as follows [4]: a): Associate a quiver node $Q_i$ with gauge group $U(n_i)$ for a component $C_i$; b): The number

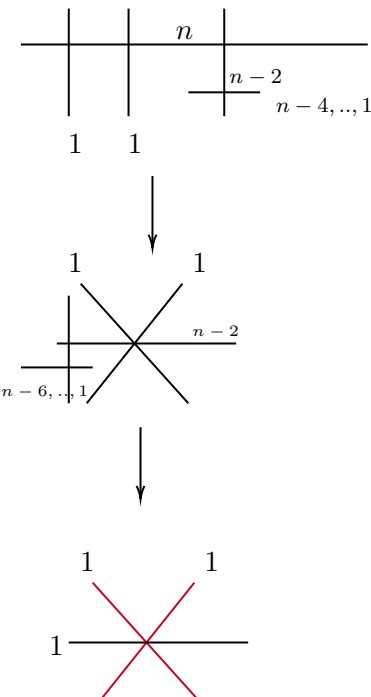

**Figure 14**: The contracting process. At the end, the intersection number between two red lines is $\frac{n-1}{2}$. The intersection number of red and black lines is just one.

of bi-fundamental fields between $Q_i$ and $Q_j$ is given as $n_{ij} = C_i \cdot C_j$; c): Add $g_i$ adjoint hypermultiplets for a node $Q_i$. The Higgs branch dimension of the quiver gauge theory is computed as:

$$
\begin{aligned}
r_H &= \sum_{i<j} n_i n_j C_i \cdot C_j + \sum g_i n_i^2 - \sum n_i^2 + 1 \\
&= \frac{1}{2}(\sum C_i n_i)^2 + \sum g_i n_i^2 - \frac{1}{2} C_i^2 n_i^2 - \sum n_i^2 + 1 \\
&= \sum n_i^2 (g_i - \frac{1}{2} C_i^2 - 1) + 1 \\
&= \boxed{\sum \frac{(n_i^2 - n_i)}{2} k_i} + g.
\end{aligned}
\tag{4.3}
$$

Here $F^2 = (\sum n_i C_i)^2 = 0$ is used, and an extra $+1$ is added because an overall $U(1)$ gauge group is decoupled.

Let's extract a 3d mirror for the IR theory from above quiver gauge theory. The basic match is that the above dimension formula $r_H$ should be equal to $g$ (which is the Coulomb branch dimension of the 4d IR theory), but in general the above formula is larger than $g$. It was noticed in [4] that one can match the known 3d mirror by modifying the proposed quiver gauge theory, and it is possible to generalize the construction in [4] to theory of arbitrary rank (again by matching the known 3d mirror for 4d theories). Here is the summary for the modification procedure:

1. If a quiver node has $n_i = 1$ or $k_i = 0$ (this is only possible for $-2$ rational curve, see formula. 4.1 by using $C_i^2 < 0$), then no modification is needed, as these nodes do not contribute to the extra terms in the Higgs branch dimension formula. 4.3.

2. For other nodes, one need to peel off $\boxed{k_i}$ number of following quiver tails

$$n_i - (n_i - 1) - (n_i - 2) - \ldots - 2 - 1$$

   Each such tail contributes Higgs branch dimension $\frac{(n_i^2 - n_i)}{2}$, and so would cancel the contribution of the $i$th node in the formula. 4.3. Notice that this is not always possible.

*Example*: Let's consider the periodic map given by the data $(n = 4, g' = 0, \frac{3}{4} + \frac{3}{4} + \frac{3}{4} + \frac{3}{4})$. The quiver gauge theory derived from it is a star-shaped quiver with four maximal quiver tails. The central node has self-intersection number $C_i^2 = -3$, and has $k_i = 1$, so one need to peel off one maximal quiver tail. The final quiver is the 3d mirror for $T_4$ theory [21]. More evidence for the identification would be given later.

The rule listed above is based on the match between the 3d mirrors for known SCFTs. It would be interesting to find a deeper reason for it, and perhaps those bad ones do not appear as the singular fiber for the SW geometry. Indeed, one can not find physical interpretation if above constraints are not imposed. Here let's illustrate this point by an interesting example. Consider a genus two example, and the weighted graph is $(g = 2, r = 1)$, and the cut is amphidrome. The valency data is $(\frac{1}{2}) + \frac{3}{4} + \frac{3}{4}$ (the one in the bracket is for the cutting curve) (this is the $III^* - II_n^*$ singularity in [4]). Now the conjugacy class has an integral parameter $n$, whose physical interpretation should be a gauge theory coupled with free hypermultiplets (see section. 4.4). However, according to our dimension formula (see table. 4 in [4]), there are only two possible scaling dimensions $(2, \frac{3}{2})$ and $(4, 3)$ for this degeneration. The scaling dimension $(2, \frac{3}{2})$ gives the gauge theory description for the fiber $(III - II_n^*)$, so the scaling dimension for fiber $III^* - II_n^*$ should be $(4, 3)$ which implies that no gauge theory is available (there has to be a dimension two Coulomb branch operator so that a gauge theory description is possible). This seemingly contradiction is saved by our constraints from finding a 3d mirror. In fact, one only find a good 3d mirror by setting $n = 0$, and interestingly the scaling dimension read from the 3d mirror is indeed $(4, 3)$. The above example gives certain justification to our procedure, and more evidence would be given later, see section. 4.4.

**Bad tail**: Let's now analyze a general tail determined by the fraction $\frac{a}{n}$ with the order $mn$, here $(a, n) = 1$. Let's assume following decomposition of $n$:

$$n = (n - a)x + b, \quad 0 < b \leq (n - a)$$

The quiver tail takes the following form (for $a \geq \frac{n}{2}$):

$$n - (n - (n - a)) - (n - 2(n - a)) - \ldots - (n - (x - 1)(n - a)) - \boxed{b} - \ldots$$

The node with multiplicity $b$ is a bad node (its self-intersection number is not $-2$). For $a < \frac{n}{2}$, the quiver tail is

$$n - \boxed{a} - c - \ldots$$

Here the node with multiplicity $a$ is a bad node.

Let's now analyze the constraint on the data $(a, n)$ so that a good tail can be defined using our modification procedure.

- For $m = 1$: Let's first assume $a \geq \frac{n}{2}$: 1): the tail has to take the form $\ldots - (n - (x-1)(n-a)) - \boxed{b} - (b-1) - \ldots - 1$. To implement our modification procedure, the self-intersection number of node $b$ has to be $-3$:

$$3b = (b-1) + (n - (x-1)(n-a)) \rightarrow b = n - a - 1.$$

2): If $b = 1$, then no problem arises. Similarly, if $a < \frac{n}{2}$, the tail has to take the form $n - a - (a-1) - \ldots$ and $a$ should have self-intersection number $-3$, which gives $a = \frac{n-1}{2}$. The above constraints on $(n, a)$ can be put in following uniform way:

$$n = (n-a)x + (n-a-1) \quad or$$
$$n = (n-a)x + 1 \tag{4.4}$$

One can associate a quiver tail from a Young Tableaux [21]. The Young Tableaux for above cases are shown in figure. 15.

- For $m > 1$, one can not use the modification procedure to cure the bad node, and so all the node has to be good. This has only following two solution: $a = n - 1$ or $n = 1$. The associated Young Tableaux is listed in figure. 15.

- Finally, let's point out that one must be careful if the dual graph has $-1$ curve as the core, and one need to first contract the $-1$ curves and then decide whether the dual graph is good or not. A case by case study is needed.

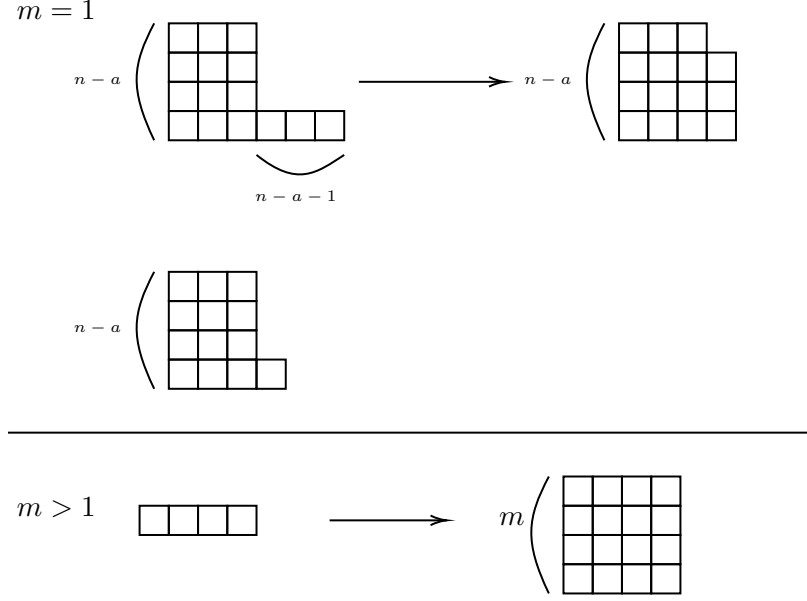

**Figure 15**: Up: The Young Tableaux for $m = 1$ which would give the good tail, and the flavor symmetry is $SU(x) \times U(1)$ (after modifying the quiver tail), here $n = (n - a)x + (n - a - 1)$ or $n = (n - a)x + 1$; Bottom: The Young Tableaux for $m > 1$, and the flavor symmetry is $SU(\lambda)$, with $\lambda m = n$.

## 4.2 Periodic map and 4d SCFTs

It is natural to associate a 4d $\mathcal{N} = 2$ SCFT for the periodic map, see [12] for rank one and [4] for rank two cases. Here let's do it for theory with arbitrary rank, see table. 1 for many interesting class of examples, and the main check is the relation between dual graph and 3d mirror. Lower genus cases are studied in table. [2,3,4,5,6], and the corresponding physical theory is studied in table. [35,36]. Let's now point out some interesting relations between geometric data and the physical data.

**Maximal scaling dimension and order of period map**: There is a relation between the order of periodic map and the maximal scaling dimension in Coulomb branch spectrum: Assume the common divisor of the scaling dimension is $u$, and the maximal scaling dimension takes the form $u_{max} = \frac{n}{u}$, then the order of periodic map is $n$. The reason is the relation between the scaling dimension and singularity spectrum [25]. In particular, the maximal scaling dimension is given as

$$u_{max} = \frac{1}{\alpha_0};$$

Here $\lambda_0 = \exp^{2\pi i \alpha_0}$ with $\lambda_0$ the eigenvalue of the monodromy group, whose order is the same as that of the periodic map, i.e. $\lambda_0^n = 1$. This implies $\alpha_0 n = 1 (mod\ \mathbb{Z})$, which gives the desired relation.

*Example*: Let's consider the possible periodic map associated with the $(A_{n-1}, A_{k-1})$ theory. The maximal scaling dimension of this theory is $\frac{nk}{n+k}$, and so the order of the

periodic map should be $nk$, and indeed this is the case, see table. 1 for examples.

**Generic deformation and number of $A_1$ singularities**: Let's verify our proposal by looking at the $T_n$ theory, which is given by an order $n$ periodic map, see table. 1 for the valency data. An interesting number is the number of $A_1$ singularities under the generic deformation. These numbers can be computed as follows: first let's compute central charge $c$, and then use the following formulas [26]:

$$c = \frac{R(B)}{3} + \frac{r}{6}, \quad R(B) = \frac{\alpha_{max}\mu}{4}, \quad \to \quad \mu = \frac{12c - 2r}{\alpha_{max}}.$$

Here $\mu$ is the number of $A_1$ singularities, $\alpha_{max}$ is the maximal scaling dimension in the Coulomb branch spectrum, $r$ is the rank of theory. Since $c = \frac{n^3}{6} - \frac{n^2}{4} - \frac{n}{12} + \frac{1}{6}, r = \frac{(n-1)(n-2)}{2}$, $\alpha_{max} = n$ for $T_n$ theory [27], one get:

$$\mu = 2(n-1)^2.$$

Now for the periodic map which is used to engineer $T_n$ theory, it admits deformation so that all the singularities are the $A_1$ singularities [28], and the number is:

$$\#A_1 = n_t + 2g - 1;$$

Here $n_t$ is the number of components in the dual graph of the periodic map, and is equal to $n_t = n(n-1) + 1$. Since $g = \frac{(n-1)(n-2)}{2}$ which gives the rank $r$ of theory, and finally $\#A_1 = 2(n-1)^2$ from above formula. This is in agreement with the computation using central charge. For the theories engineered using isolated three dimensional singularities, the number of $A_1$ singularities is equal to $2r + f$ where $f$ is the number of mass parameters. This relation does not hold in general, as the example with $T_n$ theory shows.

**Inverse engineering**: Since the 3d mirror for a large class of 4d $\mathcal{N} = 2$ theories are known [7, 21, 23], one can try to get the periodic map by doing inverse enginneering. The idea is following: a) if the 3d mirror is a star-shaped quiver with central node $n$, and the quiver tail is constrained so that it can be derived from a ratio $\frac{a_i}{\lambda_i}$ ($m_i\lambda_i = n$), see section. 3.1; then one need to add $x$ maximal tails so that the new quiver can be described by the dual graph of a periodic map. $x$ is determined by following equations:

$$N = \frac{x(n-1) + \sum m_i a_i}{n} \in \mathbb{Z},$$
$$N - 2 = x.$$

The first equation ensures the valency data satisfies integral equation for the periodic map, and the second equation is the condition for peeling off $x$ extra maximal tails. In conclusion, one solve $x$ to be

$$x = \sum m_i a_i - 2n. \tag{4.5}$$

The valency data is then $(n, g' = 0, \sum \frac{a_i}{\lambda_i} + x\frac{n-1}{n})$.

b): The 3d mirror for Argyres-Douglas is more complicated, and it is often given by a complicated quiver core (which consists more than one quiver nodes, such as a triangle,

| Theory | Data |
|---|---|
| $SU(n)$ with $n_f = 2n$ | $g = n - 1$, $(n, g' = 0, \frac{1}{n} + \frac{1}{n} + \frac{n-1}{n} + \frac{n-1}{n})$ |
| $SU(2)^k$ linear quiver | $g = k$, $(n = 2, g' = 0, (\frac{1}{2})^{2k+2})$ |
| $T_n$ | $g = \frac{(n-2)(n-1)}{2}$, $(n, g' = 0, \underbrace{\frac{n-1}{n} + \frac{n-1}{n} + \ldots + \frac{n-1}{n}}_{n})$ |
| $(A_1, A_{n-1})$, $n$ even | $g = \frac{n-2}{2}$, $(n, g' = 0, \frac{1}{n} + \frac{1}{n} + \frac{n-2}{n})$ |
| $(A_1, A_{n-1})$, $n$ odd | $g = \frac{n-1}{2}$, $(n, g' = 0, \frac{1}{2n} + \frac{1}{2} + \frac{n-1}{2n})$ |
| $(A_1, D_{n+1})$, $n$ even | $g = \frac{n}{2}$, $(n, g' = 0, \frac{1}{2n} + \frac{1}{2} + \frac{n-1}{2n})$ |
| $(A_1, D_{n+1})$, $n$ odd | $g = \frac{n-1}{2}$, $(n, g' = 0, \frac{1}{n} + \frac{1}{n} + \frac{n-2}{n})$ |
| $(A_2, A_{3n-1})$ | $g = 3n - 2$, $(3n, g' = 0, \frac{1}{3n} + \frac{1}{3n} + \frac{1}{3n} + \frac{n-1}{n})$ |
| $(A_{k-1}, A_{nk-1})$ | $g = \frac{(k-1)(nk-2)}{2}$, $(nk, g' = 0, \underbrace{\frac{1}{nk} + \ldots + \frac{1}{nk}}_{k} + \frac{n-1}{n})$ |
| $(A_{n-1}, A_{k-1})$, $(n, k) = 1$ | $g = \frac{(n-1)(k-1)}{2}$, $(nk, g' = 0, \frac{1}{nk} + \frac{a}{n} + \frac{b}{k})$ $(n = 3, k = 3x + 1 \to a = 2, b = x)$ $(n = 3, k = 3x + 2 \to a = 1, b = 2x + 1)$ |
| $D_2 SU(2n + 1)$ | $g = n$, $(2n + 1, g' = 0, \frac{1}{2n+1} + \frac{n}{2n+1} + \frac{n}{2n+1})$ |
| $D_{n+k}(SU(n))$, $(n, k) = 1$ | $g = \frac{(n-1)(n+k-1)}{2}$, $(n(n + k - 1), g' = 0, \frac{1}{n(n+k-1)} + \frac{a}{n} + \frac{b}{n(n+k-1)})$ $n = 3, n + k = 3x + 1 \to a = 2, b = n + k - 2$, $n = 3, n + k = 3x + 2 \to a = 1, b = 2(n + k) - 3$ |

**Table 1**: Periodic maps and the associated SCFTs.

see [7] for details), and there are also tails attached to the core quiver nodes; To find the periodic map, one need to do the inverse operation of contraction to get a star-shaped quiver.

*Example 1*: Let's consider $A_2$ class $\mathcal{S}$ theory which is engineered by a sphere with $a$ simple punctures and $b$ full punctures [6]. The 3d mirror is a star-shaped quiver with $a$ simple tail $3 - 1$, and $b$ maximal tail $3 - 2 - 1$ [21]. According to our formula 4.5, $x = a + 2b - 6$, and the valency data is

$$(\frac{1}{3})^a + (\frac{2}{3})^b + (\frac{2}{3})^{a+2b-6}.$$

A simple check is following: the rank of the SCFT is $2a + 3b - 8$. The genus formula from the valency data is

$$2g - 2 = 3(-2 + \frac{2}{3}(2a + 3b - 6)) \to g = 2a + 3b - 8,$$

which is consistent. The above computation can be done similarly for $A_{n-1}$ theory defined by a sphere with $a$ simple and $b$ full punctures.

*Example 2*: Consider an AD theory whose 3d mirror has a triangle core with ranks $n_1, n_2, n_3$. To get a star-shaped quiver whose contraction will give rise to the 3d mirror for

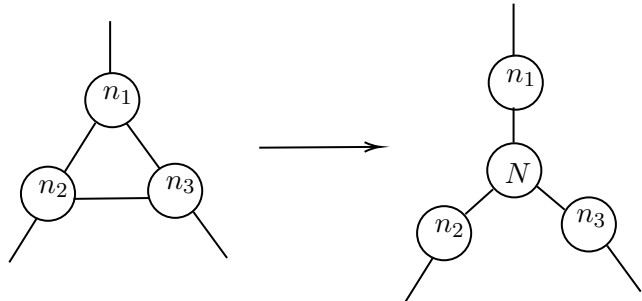

**Figure 16**: Left: the 3d mirror for an AD theory whose core has three nodes and are connected by a single edge; Right: One add a node with rank $N = n_1 + n_2 + n_3$ so it becomes a star-shaped quiver.

AD theory, one add a core node with rank $N = n_1 + n_2 + n_3$, see figure. 16. The valency data is

$$\frac{n_1}{N} + \frac{n_2}{N} + \frac{n_3}{N}$$

$n_1, n_2, n_3$ is chosen that the above data is legitimate to define a periodic map, and moreover they are constrained so that one can use the modification procedure to get a 3d mirror. Some solutions are $n_1 = 1, n_2 = n, n_3 = n$ (one get 3d mirror for $D_2 SU(2n + 1)$ theory), and $n_1 = n + 1, n_2 = n, n_3 = n$ (here $n \leq 4$).

### 4.3 Other 4d SCFTs

It was noticed in [4] that many familiar rank two 4d $\mathcal{N} = 2$ SCFTs are not given by the periodic maps. The reason is following: our study is based on one parameter family of genus $g$ fiberation, but one can not find a one parameter scale invariant geometry for many SCFTs, therefore a periodic map of the genus $g$ can not be associated. However, it seems possible to find a non-scale invariant one parameter genus $g$ family associated with such family, which fits in our framework. So it should be possible to discover those SCFTs using our methods. More importantly, one always need to put a genus $g$ degeneration at $\infty$ [3], and so it is important to identify the conjugacy classes of these SCFTs. Here are some possibilities:

1. First, let's consider rank $l$ version of $E_n$ [29] and $H_n$ [30] type theories. The corresponding weighted graph is shown in figure. 17. One might wonder whether it is possible to find other similar rank $l$ generalizations, namely, the weighted graph is similar to that shown in figure. 17, and with the replacement of genus one component by a genus $g$ component. The search turns out to be difficult, as there are strong constraints on original theory: a) no modification is needed for original dual graph; b) all the valency data should be of the form $\frac{n-1}{n}$. Most likely, one can only consider $E_n$ and $H_n$ theories.

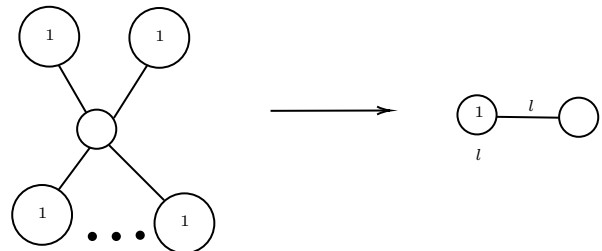

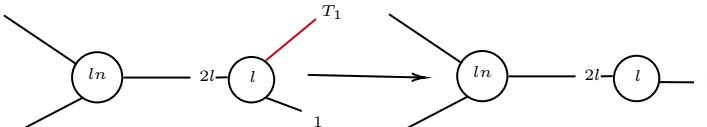

**Figure 17**: Up: The weighted graph corresponding to rank $l$ version of $H_n$ and $E_n$ theories, and the automorphism permutes the genus one nodes. The periodic map on genus zero component is $\mathbf{1} + \frac{1}{l} + \frac{l-1}{l}$, and on the genus one component is $\frac{3}{4} + \frac{3}{4} + \frac{1}{2}$ (we take $E_7$ for an example, and other cases are similar). Bottom: the dual graph for above configuration, here $K$ (the integer associated with the cut curves) is take to be zero; After modification on the bad node, one get the 3d mirror for rank $l$ theory.

2. One might also get SCFT from the geometry $(g, r = 1)$ with the cut curve amphidrome. The valency data for the cut curve $C_1, C_1'$ is $\frac{n-1}{n}$, and the order is $2n$.

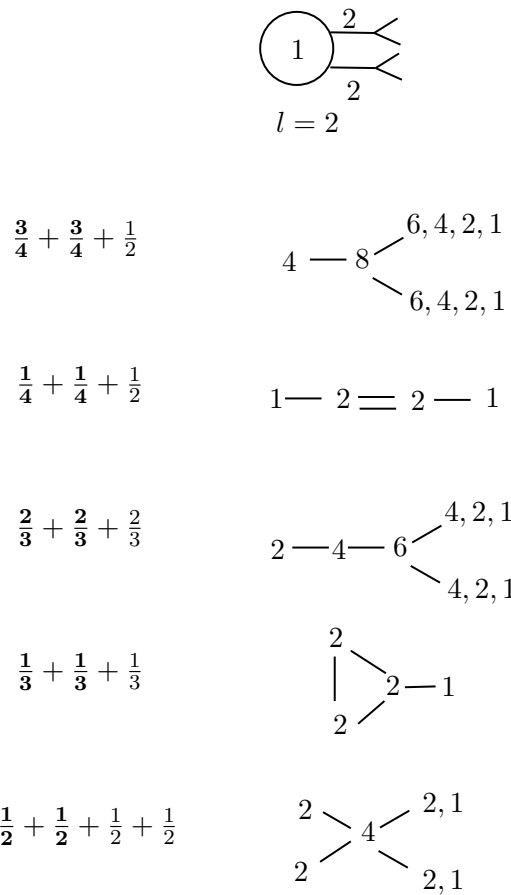

**Figure 18**: Top: Weighted graph which might give a SCFT. The valency data for periodic map is given on the left, and the 3d mirror from the dual graph is shown on the right.

The tail for the amphidrome curve is then

$$2n - (2n - 2) - \ldots - 4 - 2 - (1,1)(terminal)$$

One can take off one terminal node with rank one to get a good 3d mirror. So the valency data is

$$(\frac{n-1}{n}) + \sum \frac{a_i}{2n}$$

Here $a_i$ is chosen so that one has a good tail. This will produce the 3d mirror for many known SCFT.

3. There are some other possibilities: the weighted graph has one component and the edges are all of the $D$ type. We leave the detailed analysis to interested reader. Here we give a class of examples.

*Example*: The weighted graph is given in figure. 18. It has just one node with $l = 2$, and there are two D type tails. The corresponding data is $(g = 1, r = 0, k = 2)$, and

the periodic map fixes each boundary map separately. There are following choices for the periodic map

$$\mathbf{\frac{3}{4}} + \mathbf{\frac{3}{4}} + \frac{1}{2}, \quad \mathbf{\frac{1}{4}} + \mathbf{\frac{1}{4}} + \frac{1}{2}, \quad \mathbf{\frac{2}{3}} + \mathbf{\frac{2}{3}} + \frac{2}{3}, \quad \mathbf{\frac{1}{3}} + \mathbf{\frac{1}{3}} + \frac{1}{3}, \quad \mathbf{\frac{1}{3}} + \mathbf{\frac{1}{3}} + \frac{1}{3}$$

The corresponding dual graph (after the modification and so is the 3d mirror) is shown in figure. 18. It is easy to identify them as the rank three theories listed in table 10 of [31].

4. Finally, it might happen that the gluing parameter $K$ takes value $K \geq 0$, and only $K = 0$ can be consistently to find a 3d mirror (this happens for $m > 1$). This often implies that one can associate a SCFT for it.

*Example*: Let's give a family of examples to illustrate this class of examples. The weighted graph is given by two components (with genus $g_1$ and $g_2$ separately) connected by an edge with weight 2, see figure. 19. The valency data are

$$\mathbf{\frac{1}{3n+4}} + \frac{1}{6n+8} + \frac{6n+5}{6n+8}, \quad \mathbf{1} + (\frac{1}{2})^{2n+2}$$

The dual graph gives (after modification) a good 3d mirror if $K = 0$. In fact, the 3d mirror agrees with the theory found in [7]: it is engineered by an irregular singularity with data $A_2$ theory on a sphere with one irregular singularity with data $\Phi = \frac{T_{2n+4}}{z^{2n+4}} + \ldots + \frac{T_2}{z^2} + \frac{T_1}{z}$, with $T_{2n+4}, \ldots, T_2$ are of type $[2,1]$, and $T_1$ are type $[1,1,1]$, there is also a maximal regular singularity.

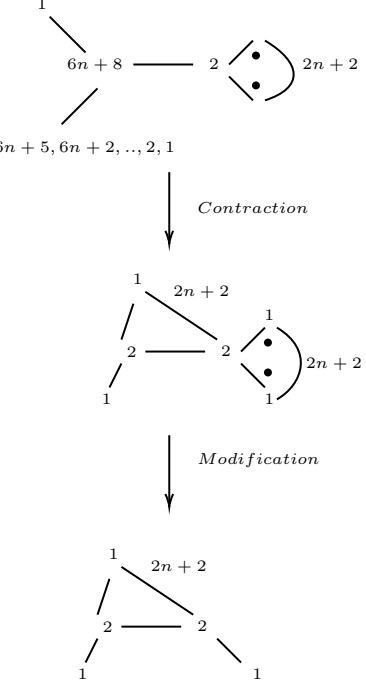

**Figure 19**: Up: the dual graph for a pseudo-periodic map, and the weighted graph has one genus $g_1$ component connected by a genus zero component. Bottom: the corresponding 3d mirror.

### 4.4 IR free theory

Let's now consider IR theory associated with general pseudo-periodic map. It is expected that the IR theory is given by gauge theory coupled with matter. Roughly, the matter is represented by irreducible components in the cut system, and the gauge group is represented by edges (the rank of the gauge group is given by the weights of the edge), and the number $K$ for the edges gives the hypermultiplets coupled with gauge group.

To begin with, let's analyze the case where the weighted graph has only one node, so the data is $(g, r)$, here $r$ denotes the non-separating cutting curves. One can find the IR theory (with arbitrary $K$) as follows:

1. Let's take $r = 1$ and the cut non-amphidrome, and $f$ be the identity map on $\Sigma_g/C$, the low energy theory is $U(1)$ gauge theory with $K$ fundamental hypermultiplets and $g - 1$ free vector multiplets. This can be seen from the dual graph (and 3d mirror) as follows: the 3d mirror consists $g - 1$ free hypermultiplets [7] which gives $g - 1$ free vector multiplets of the original theory, and a cyclic quiver with $K$ $U(1)$ gauge nodes whose mirror is just $U(1)$ with $K$ hypermultiplets [32], see figure. 20.

2. Again, let's take $r = 1$ and the cut curve non-amphidrome, but $f$ is now a general periodic map. Now the low energy theory can be found as follows: Let's assume the

---

[7]The adjoint hypermultiplets on $U(1)$ node decouple.

boundary curves are $C_1$ and $C_2$, and the theory associated with $\Sigma_{g-1}$ is $T$, which gives a rank $g-1$ theory. The dual graph has tail $T_1$ and $T_2$ corresponding to boundary curves $C_1, C_2$ (The valency data for them has $m = 1$). The low energy theory is then described as follows: it is given by diagonally gauging $U(1)$ flavor symmetry of the tail $T_1$ and $T_2$, and there is also $K$ extra free hypermultiplets charged over $U(1)$ gauge group. Since the modification procedure is no longer available (as the two tails have to be glued together), one has following constraint on the valency data $(n, a)$: $n = (n-a)x + 1$, and examples are $\frac{n-1}{n}$ or $\frac{1}{n}$.

3. Let's take $r = 1$, but this time the action is amphidrome. The boundary curve $C_1'$ and $C_1''$ now are related by the periodic map (the valency data for them has $m = 2$). And the dual graph has a $D$ type tail. To get a valid 3d mirror with arbitrary link number $K$, the order of periodic map should be two, and the valency data for the cutting curves are $\mathbf{1}$, see figure. 20. We interpret it as a $SU(2)$ gauge theory coupled with $K + 2$ fundamental flavors and a theory $T$ represented by the periodic map on $\Sigma_{g-1}$. One of the evidence is that by increasing $K$ by 1, the Coulomb branch dimension of the 3d mirror is increased by two, and so the Higgs branch of the original theory is increased by two. This is consistent with the interpretation that $K$ is the number of fundamental hypermultiplets coupled with $SU(2)$ gauge group.

4. The situation with general $r$ is now clear: a): if the action is non-amphidrome, and one of the orbit of $f$ is $\langle C_1, \ldots, C_s \rangle$, the boundary curves after the cut are $(C_1', \ldots, C_s'), (C_1'', \ldots, C_s'')$, then there are two tails for the cutting curves $(C_1', \ldots, C_s')$, $(C_1'', \ldots, C_s'')$: the low energy theory is just gauging diagonally $\boxed{U(s)}$ flavor symmetry of $T_1$ and $T_2$, and the valency data for them is $\mathbf{1}$ ; b): if the action is amphidrome, and the orbit of $f$ is $\langle C_1, \ldots, C_s \rangle$, the boundary curves after the cut are $(C_1', \ldots, C_s'), (C_1'', \ldots, C_s'')$, then $(C_1', \ldots, C_s', C_1'', \ldots, C_s'')$ are under the same orbit of $f$ (the valency data is $\mathbf{1}$), and the gauge group is conjectured to be $\boxed{Sp(2s)}$. An example is shown in figure. 20.

Let's now discuss the situation where weighted graph has several components. Now each component has the label $(g_i, r_i, k_i)$ with $k_i$ the boundary curves connecting with other components. The physical interpretation for the non-separating curves $r_i$ are gauging, which is also true for separating curves. The difference for the separating curves is that the gauge group is $SU(m)$ (instead of $U(m)$ ). This is due to the fact the contribution of the edge with multiplicities $m$ to the genus is $m - 1$. There are also $K$ fundamental hypermultiplets charged over $SU(m)$ gauge group. In particular, if $m = 1$, there are decoupled massless hypermultiplets (as the gauge group is $SU(1)$).

So the IR theory for a general pseudo-periodic map is given by gauging the matter systems represented by periodic maps. Some familiar matter systems are given:

1. **Free matter**: A set of free matter is represented by a sphere with $n$ boundary curves, which are permuted by the action $f$. The periodic map is represented by the data $\mathbf{1} + \frac{1}{n} + \frac{n-1}{n}$, which gives rise to bi-fundamental matter.

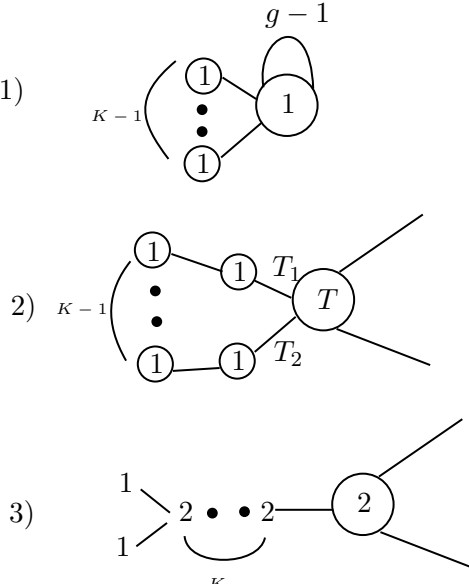

**Figure 20**: (1): Dual graph for $(g, r = 1)$: the cut curve is non-amphidrome, and the corresponding periodic map is trivial; The IR theory is $U(1)$ coupled with $K$ free hypermultiplets and $g - 1$ free vector multiplets; (2): Dual graph for $(g, r = 1)$: the cut curve is non-amphidrome and the corresponding periodic map is generic; (3): Dual graph for $(g, r = 1)$, the cut curve is amphidrome and the corresponding periodic map has order two. The IR theory is $SU(2)$ gauge theory coupled with $K + 2$ free hypermultiplets and an interacting theory.

More generally, one can consider a sphere with $2n$ boundary curves, and $f$ permutes $n$ components separately. The periodic map is given by the data $\mathbf{1} + \mathbf{1} + \frac{1}{n} + \frac{n-1}{n}$. The dual graph is shown in figure. 21. The self-intersection number for the core component is $-3$, and so one need to take off a maximal quiver tail, and the 3d mirror is shown in figure. 21. It is then easy to see that this corresponds to the mirror of bi-fundamental matter with flavor group $SU(n) \times SU(n) \times U(1)$.

2. $T_n$ **matter**: It is represented by a genus $\frac{(n-1)(n-2)}{2}$ curve with $3n$ boundary components, with $f$ permuting $n$ components each. The periodic map is given by the data $\mathbf{1} + \mathbf{1} + \mathbf{1} + \underbrace{\frac{n-1}{n} + \ldots + \frac{n-1}{n}}_{n}$. The dual graph and the corresponding 3d mirror are shown in figure. 21. It is then natural to identify it as the $T_n$ matter, with three $SU(n)$ flavor symmetry groups gauged.

3. **General matter**: It is now clear how to get general matter system. Let's start with a periodic map with order $n$, so that it can be interpreted as a SCFT. Now assume that there is still maximal tail left (with the valency data $\frac{n-1}{n}$) in the 3d mirror, then one can consider the following configuration: a genus $g$ curve with $n$ boundary curves which are permuted by periodic map. Then the data for the periodic map

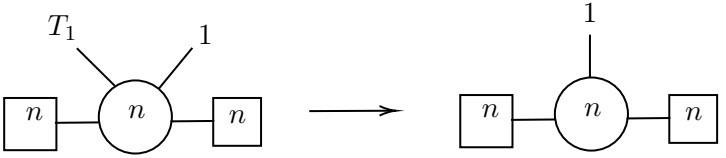

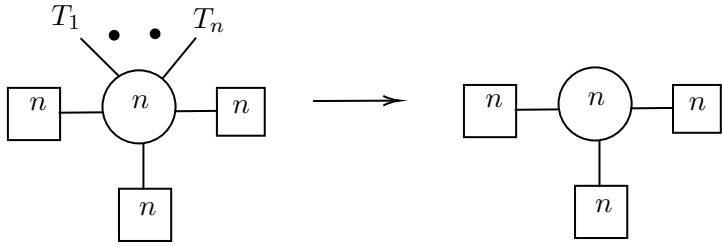

**Figure 21**: Up: the dual graph for bi-fundamental matter, and after modification it becomes a good 3d mirror (part of a larger quiver); Bottom: the dual graph for $T_n$ matter, and after modification it becomes a good 3d mirror. Here $T_i$ denotes the quiver tail corresponding to maximal regular puncture.

of this configuration is $\mathbf{1} + \ldots$, where $\mathbf{1}$ represents the $n$ boundary curves. To get the good dual graph, one simply remove a maximal tail for the modified dual graph, and replace it with a tail with one node $\boxed{n}$. The gluing of this matter system with other parts can be interpreted physically as gauging the $SU(n)$ flavor symmetry of the original matter system.

4. **Special case**: The following matter system deserves special treatment: The periodic map is given by $\sum \frac{\sigma_i}{\lambda_i} + \mathbf{1}$, here $\sum \frac{\sigma_i}{\lambda_i} = 1$. The order is $n$, and there are $n$ boundary curves which are permuted by the period map. The matter system is interpreted as the class $S$ theory with regular punctured labeled by $\frac{\sigma_i}{\lambda_i}$, and a full regular puncture. The gluing is interpreted as gauging the $SU(n)$ flavor symmetry.

Using above matter system, one can easily construct familiar IR free theory and the associated pseudo-periodic map, see figure. 22. The 3d mirror for such IR free theory has been found in [22].

**Constraints on gluing**: Important rule on gluing is that the dual graph can be interpreted as the 3d mirror for the low energy theory (with possible modification). This puts constraints on possible gluing pattern, and a detailed analysis will be provided.

Let's look at the formula in section. 3.2 for gluing dual graphs, and one can make following observations. If the curves $C_i$ are non-amphidrome, then

1. If $m = 1$, since it is no longer possible to do the surgery to modify the quiver tails, the two tails $T_1, T_2$ have to be good by itself (all the quiver nodes on the tail has

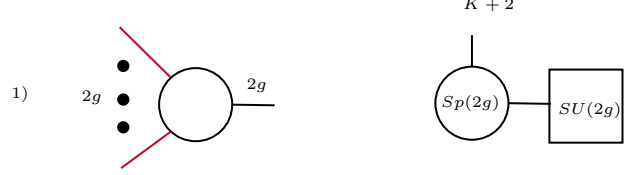

1)

2)

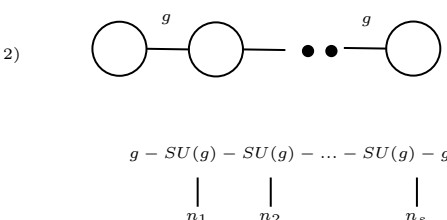

$$g - SU(g) - SU(g) - \ldots - SU(g) - g$$

$$\underset{n_1}{\big|} \qquad \underset{n_2}{\big|} \qquad \underset{n_s}{\big|}$$

3)

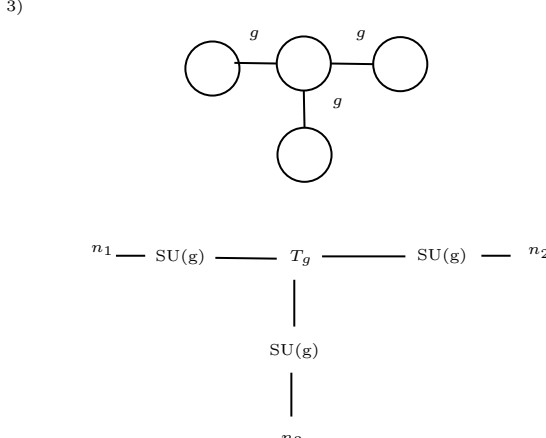

$$n_1 - SU(g) - T_g - SU(g) - n_2$$

$$\big|$$

$$SU(g)$$

$$\big|$$

$$n_3$$

**Figure 22**: Some IR free gauge theories.

self-intersection number $-2$ and no modification is needed). This puts the constraint on valency data: it can only take the form $\frac{a}{n}$, with $n = (n-a)x + 1$. The glued tail would be

$$(n_1 - \ldots - 1 - \underbrace{1 - \ldots - 1}_{K-1} - 1 - \ldots - n_2), \quad K \geq 1$$

$$(n_1 - \ldots - 1 - 1 - \ldots - n_2), \quad K = 0$$

For $K = -1$, one need to verify the condition case by case.

2. If $m > 1$, the only solution would be

$$(m - \underbrace{m - \ldots - m}_{K-1} - m)$$

then there would be no bad component; the boundary valency data is actually $(\mathbf{1}, \mathbf{1})$, and $K \geq 1$;

If the curve $C_i$ is amphidrome (the valency data for the cutting curves are $\frac{a}{\lambda}$ with $n = 2m\lambda$), by looking at the glued quiver, one find that:

1. $2m = n$ ($n$ the order of the periodic map, and the valency data is $(1)$), then there would be no bad node as the tail takes the form

$$2m - \underbrace{2m - \ldots - 2m}_{K} - (m, m)(\text{terminal node})$$

2. $m = 1$, $K = 0$, and $a_{u-1} = 2$ *or* $3$. Since the tail takes the form

$$\ldots - 2a_{u-1} - \mathbf{2} - (1, 1),$$

   and the bad node $\mathbf{2}$ has $k = a_{u-1} - 1$, so if $a_{u-1} = 2, 3$, one can peel off one or two maximal tails to get a good tail.

The above constraints then significantly reduce the possibilities for the gluing:

- If two components in weighted graph are connected by an edge with weight $m$, then physically it is interpreted as gauging a $SU(m)$ flavor symmetry of the matter. There are also $K$ fundamental hypermultiplet charged with $SU(m)$. In particular, if $m = 1$, there is no gauging, and the IR theory is the direct sum of the theory associated with each component; there are also $K$ free hypermultiplets which do not carry any electric-magnetic charge.

- For the cuts corresponding to non-separating curves, the gluing is interpreted as a $U(m)$ gauge group, or $Sp(2m)$ gauge group depending whether the cut is amphidrome or not. For $m > 1$, there is only $SU(m)$ flavor symmetry for the matter associated with theory defined using period map, but there is an extra $U(1)$ gauge theory, so that total rank of the IR theory is the same as the genus of the Riemann surface.

- When the periodic map of one component is special (namely the sum of valency data is one), one should be careful about the gluing, and a case by case analysis is needed. The physical interpretation is more complicated, many gauge theory coupled with two Argyres-Douglas matters can be described in this way.

- If the vertices form a loop in the weighted graph, then the gauge groups take the form $\sum SU(m_i) \oplus U(1)$. The extra $U(1)$ is needed as the contribution to the genus of the edges in the loop is $\sum(m_i - 1) + 1$. Here $m_i$ are the weights of the edges connecting the vertices in the loop.

*Example 1*: Let's verify above proposal by looking at an example. The weighted graph has only one node and the data is $(g, r = g)$, and the periodic map permutes the curve $\langle C_1, \ldots, C_g \rangle$. The valency data is $(1) + (1) + \frac{1}{g} + \frac{g-1}{g}$, and the dual graph is shown in

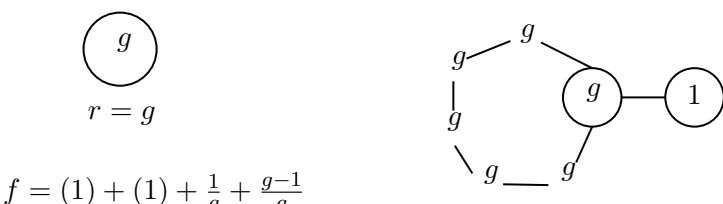

**Figure 23**: Left: The weighted graph has just one vertex and the data is $(g, r = g)$, and the periodic map data is also given. Right: The dual graph for the pseudo-periodic map.

figure. 23. It is then easy to recognize that it is the 3d mirror for $U(g)$ gauge theory with an adjoint and $K$ fundamental [33] ($K \geq 1$).

*Example 2*: The weighted graph is given by two genus one curves connected by a weight two edges, so the genus is three. The valency data on each component is $\frac{1}{4} + \frac{1}{4} + \frac{1}{2}$, and $K = -S(C) - 1/2 - 1/2 \geq 0$. The dual graph is drawn in figure. 24. One recognize that it is the 3d mirror for a gauge theory: $SU(2)$ gauge theory coupled with two $D_2(SU(3))$ matter and $K + 2$ fundamental hypermultiplets. Notice that the limit $K = -1$ is actually a SCFT, which is represented by a periodic map with data $(n = 4, g' = 0, \frac{1}{4} + \frac{1}{4} + \frac{1}{4} + \frac{1}{4})$.

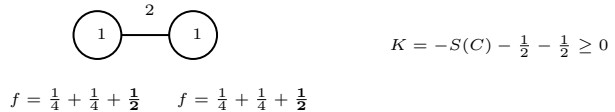

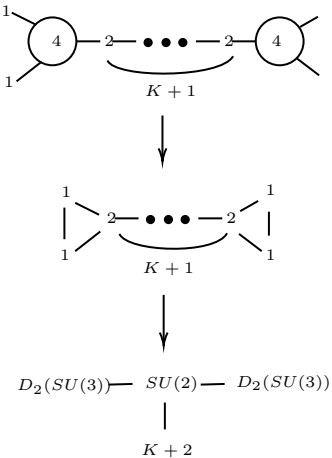

**Figure 24**: Up: the data for a pseudo-periodic map which would give the gauge theory coupled with AD matter; Middle: the dual graph and the corresponding 3d mirror; Bottom: The corresponding 4d gauge theory which is given by $SU(2)$ gauge group coupled with AD matters.

**Relation with class $\mathcal{S}$ theory**: Let's recall the geometric representation of a theory

in class $\mathcal{S}$ theory [6, 7]. The theory can be either a SCFT or an asymptotical free gauge theory. The theory is represented by a Riemann surface $\Sigma$ with regular punctures and irregular punctures, and the gauge theory is given by finding a pants decomposition of $\Sigma$. Notice that the AD matter is represented by a sphere with one irregular and one regular singularity, and the component of AD theory is living at the boundary of the pants decomposition [7]; $T_n$ matter are represented by a punctured three sphere, see figure. 25. The picture of the IR theory found above is very similar to class $\mathcal{S}$ theory: the weighted graph is similar to the pants decomposition of the Riemann surface $\Sigma_g$, and the difference is that: a): We get SCFT and IR free gauge theory; b): The matter is represented by periodic map on an irreducible component in the cut system; c): The gauge group is also represented by edges, but there is an extra integer $K$, which gives rise to free hypermultiplets coupled with gauge group. In the next section, we will show how to use our result to describe UV complete theories.

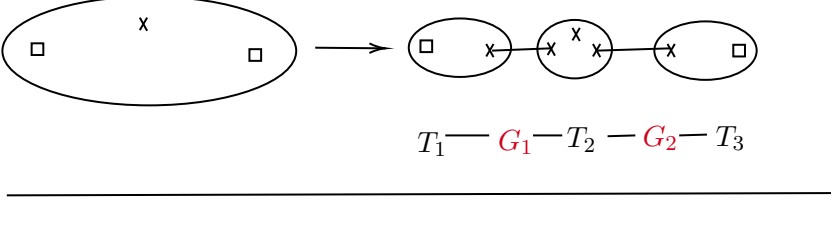

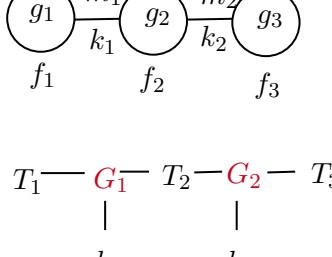

**Figure 25**: Up: The representation of a UV complete theory in class $\mathcal{S}$ description: it is given by a Riemann surface with regular and irregular singularities; and the gauge theory is given by the pants decomposition of the punctured Riemann surface and the irregular singularity stays at the boundary; Bottom: The representation of a IR free theory by a pseudo-periodic map, here one need to specify a weighted graph and the corresponding periodic map; there is an extra integer on the edges of the weighted graph.

## 5 UV theory

In last fewer sections, the IR theory associated with the degeneration is studied. In this section, we will study the global SW geometry (on a one parameter slice on the whole Coulomb branch). A point is added at $\infty$ to make the Coulomb branch compact, i.e. the Coulomb branch is now $\mathbb{P}^1$. The property of $UV$ theory is then reflected by the singular fiber at $\infty$ [3], see figure. 26.

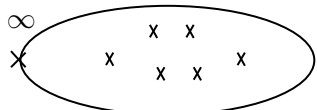

**Figure 26**: The global structure of SW geometry at Coulomb branch.

Geometrically, we now have a compact surface $S$ fibered over $\mathbb{P}^1$: $f : S \to \mathbb{P}^1$, and there are singular fibers at $\infty$ and bulk. Two further assumptions on the fiberation are made: a) First the surface $S$ is a rational surface; b): the bulk singularities are atomic, which means it can no longer be deformed. In our case, the atomic singular fiber are classified into following kinds:

1. There is a single non-separating curve. The weighted graph for it has a single node with $r = 1$ (non-amphidrome and $K = 1$), and the periodic map is trivial. The low energy theory is $U(1)$ gauge theory coupled with one massless hypermultiplet, and $g - 1$ free vector multiplets. This is denoted as $A_1$ singularity.

2. There is a single separating curve, and it separates a genus $g$ curve into a genus $h$ and a genus $g - h$ curve, see figure. 27 for the illustration. The link number is also $K = 1$, and this singularity is denoted as $A_1^h$ singularity. The low energy theory is $g$ free vector multiplets, and one free hypermultiplet (which does not charged under any low energy $U(1)$ gauge group).

In the following, we mainly consider the case where the bulk singularities are all of the $A_1$ type, and the study of general atomic singularities is left for separate publications.

**Topological invariants**: We'd like to study the global topological constraint on the singular fiber at $\infty$ for the UV theory in dimension $4, 5, 6$. Crucially, one can define two invariants $d_x, \delta_x$ for a singular fiber. $d_x$ is defined using holomorphic data, and $\delta_x$ is the topological invariant. $d_x$ is defined for genus one and genus two cases [34], and it might be defined for arbitrary genus. However, $d_x$ is in general not easy to compute, and details will be discussed elsewhere. On the contrary, the topological invariant $\delta_x$ can be computed from the dual graph (the normally minimal model). In general, $d_x$ is not equal to $\delta_x$ (Though they are equal for a large class of singular fibers). Given a singular fiber $F$ with its normally minimal model, first one one get a reduced curve $F_{red}$, and its topological Euler number $\delta_x$ is computed from $F_{red}$ as follows [35]:

$$\delta_x = 2(g - p_a(F_{red})) + \#intersections \tag{5.1}$$

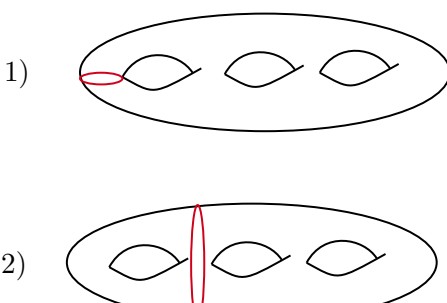

**Figure 27**: Up: The atomic singularity corresponding to non-separating curve; Bottom: The atomic singularity corresponding to separating curve.

here $p_a$ denotes the geometric genus, which is given as $p_a(F_{red}) = 1 + \frac{1}{2}(F_{red}^2 + F_{red} \cdot K)$. Let's now examine the behavior of $p_a(F_{red})$, which will have interesting implications for the UV singular fiber. Given $F_{red} = \sum_{i=1}^{k} C_i$, then

$$p_a(F_{red}) = 1 + \frac{1}{2}(F_{red}^2 + F_{red} \cdot K) = \sum_i [1 + \frac{1}{2}(C_i^2 + K_i)] + \sum_{i<j} C_i \cdot C_j - k + 1$$

$$= \sum_i (p_a(C_i)) + \sum_{i<j} C_i \cdot C_j - k + 1.$$

Since $p_a(C_i) \geq 0$ and $\sum_{i<j} C_i \cdot C_j \geq k - 1$, and the equality holds when: a) all the curves $C_i$ is a rational curve; b): all the curves intersect transversely and they form a **tree**!

One actually need to compute the topological Euler number for relative minimal model (no $-1$ rational curve). The way of computing is following: first one compute $\delta_x$ for normal model, and each contraction of $-1$ curve would reduce the Euler number by 1. Here are some comments on the Euler number $\delta_x$:

1. If the dual graph (normal minimal model) consists of only spheres, and it has $k$ loops, then the topological Euler number is given as

$$\delta_x = n_t + 2g - 1 - k;$$

   Here $n_t$ is the number of components in the dual graph. This relation still holds after contracting $-1$ curve.

2. If the dual graph can be regarded as the 3d mirror for a 4d theory (a tree graph and no modification is needed), the Euler number is related to the physical data as follows

$$\delta_x = f + 2g;$$

   Here $f$ is the rank of flavor symmetries. If the modification on dual graph is needed, the rank of flavor symmetry is smaller than the number $\delta_x - 2g$.

For our purpose, an invariant which is preserved under splitting of singularity is needed. Indeed, such invariant exists, and is called local signature [37]:

$$\boxed{\sigma(F) = \frac{1}{2g+1}(gh_x - (g+1)\delta_x).}$$

$h_x$ [8] is an invariant which measures the number of $A_1^h$ type singularities under deformation. $h_x = 0, \delta_x = 1$ for a $A_1$ singularity, and so $\sigma(F) = -\frac{g+1}{2g+1}$. The signature is preserved for the splitting of singularity, i.e. if $F$ is split into several singularities $F_1, \ldots, F_s$, then

$$\sigma(F) = \sum_i \sigma(F_i).$$

In particular, if $h_x = 0$ for a singular fiber $F$, then the number of $A_1$ singularities under the generic splitting is equal to $\delta_x$! On the other hand, if a singular fiber $F$ is split into only $A_1$ singularities under generic deformations, then $h_x = 0$.

Let's now consider global topological constraints for the fibered surface. Given a genus $g$ fiberation $f : S \rightarrow \mathbb{P}^1$, one can define the following relative invariants

$$K_f^2 = c_1^2(S) + 8(g-1),$$
$$e_f = c_2(S) + 4(g-1),$$
$$\chi_f = \chi(\mathcal{O}_S) + (g-1),$$
$$q_f = q(S).$$

Here $c_1(S)$ ($c_2(S)$) are the first (second) Chern class, $\mathcal{O}_S$ is structure sheaf, and $q(S)$ is the irregularity of $S$. The relative invariants can be expressed by the local data of singular fibers [36]:

$$K_f^2 = \kappa(f) + \sum_i c_1^2(F_i),$$
$$e_f = \delta(f) + \sum_i c_2(F_i),$$
$$\chi_f = \lambda(f) + \sum_i \chi(F_i).$$

Here $c_2(F)$ and $c_1(F)$ are the local Chern numbers, and $\kappa(f), \delta(f), \lambda(f)$ are modular invariants which are zero for **isotrivial** family. We are interested in rational surface which has $q = p_g = 0$, and so according to Noether's theorem: $c_2(S) + c_1^2(S) = 12$. The local data satisfies the following equation:

$$K_f^2 + e_f = 12g \rightarrow \kappa(f) + \delta(f) + \sum_i (c_1^2(F_i) + c_2(F_i)) = 12g. \tag{5.2}$$

---

[8]This number is equal to $d_x - \delta_x$ for rank one and rank two case, and we conjecture that it holds for general case.

The crucial data for the singular fiber is $\chi(F) = \frac{1}{12}(c_1^2(F_i)+c_2(F_i))$, which can be computed using the normal minimal model $F = \sum n_i C_i$:

$$\chi(F) = \frac{1}{2}(g - p_a(F_{red})) + \sum_{i<j} \chi(n_i, n_j) C_i \cdot C_j,$$

$$\chi(n_i, n_j) = 3 - \frac{(n_i, n_j)^2}{n_i n_j} - \frac{n_j}{n_i} - \frac{n_i}{n_j} \tag{5.3}$$

Here the bracket $(n_i, n_j)$ denotes the greatest common divisor for $n_i, n_j$. Notice that this invariant is not changed by blow-up or contraction, and so it is the same for relative minimal model. The above formula 5.2 has limited use for us as: a): It involves global modular invariants $\kappa(f), \delta(f), \lambda(f)$ which are not easy to compute; b): The local Euler number $\chi(F)$ is not conserved on deformation.

One can get more information if the fiberation is **hyperelliptic**, i.e. the generic fiber is a hyperelliptic surface. Now signature of surface $S$ is the just the sum of that of local singular fibers [37]:

$$\sigma(S) = \sum_i \sigma(F_i).$$

For a rational surface, the Hodge index theorem gives the signature of the surface as $(1, \rho - 1)$ with $\rho$ the Picard number, and so $\sigma(S) = 2 - \rho$. Therefore, one get following equation:

$$\boxed{2 - \rho(S) = \sum_i \sigma(F_i)}. \tag{5.4}$$

The Picard number for a rational surface is related to relative invariant as follows (using Noether's theorem):

$$\rho(S) = 8g + 2 - K_f^2.$$

and $K_f^2$ has a lower bound $4g - 4$ which implies an upper bound for $\rho(S) \le 4g + 6$.

**Flavor symmetry and Mordell-Weil lattice**: Given a higher genus fiberation of an algebraic surface, one can define a Mordell-Weil (MW) lattice [38]. This lattice is closed related to flavor symmetry of the physical theory, which is studied in rank one [39, 40] and rank two case [4]. Here we summarize some important properties of this lattice. First, the rank is equal to [38]:

$$r_{MW} = \rho - 2 - \sum(n_t - 1). \tag{5.5}$$

Here $\rho$ is the Picard number of $S$. For the rational fibered surface $S$ which is **hyperelliptic** [41], the Picard number is $\rho = 4g + 6$ for $K_f^2 = 4g - 4$, and the MW rank is

$$r_{MW} = 4g + 4 - \sum(n_t - 1). \tag{5.6}$$

The maximal MW rank is $4g + 4$ if all the singular fiber has $n_t = 1$, and the corresponding surface is constructed in [41], see figure. 28 for the dual graph for the lattice.

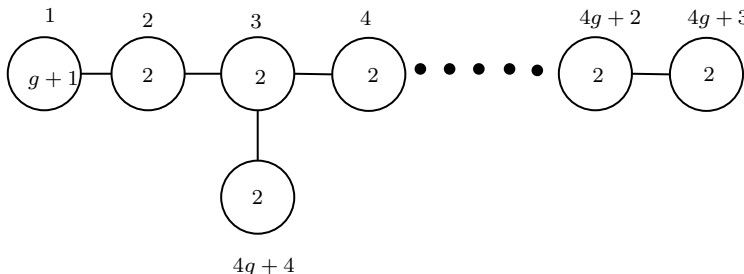

**Figure 28**: The maximal Mordell-Weil lattice for a rational surface with genus $g$ fiberation. The first node has self-intersection number $g + 1$, and other node has self-intersection number 2.

## 5.1 4d theory

Let's make following assumptions for the generic deformation of a 4d theory: a): there is one special singular fiber at $z = \infty$; b) there are only $A_1$ singularities at the bulk. We'd like to determine the topological constraint on the fiber at $\infty$ so that it can define a sensible UV theory in dimension 4.

Let's first assume that the fiberation is hyperelliptic and the Picard number is $\rho(S) = 4g + 6$. Following the same argument in [4], the topological constraint for $F_\infty$ so that it defines a 4d theory is

$$\boxed{\delta_\infty - n_\infty = 2g - 1}. \tag{5.7}$$

Let's recall the argument leading to above equation. The basic assumption is that there is one BPS particle associated with each bulk $A_1$ singularity, and they generate the whole charge lattice [9], and so

$$f + 2g = \#A_1; \tag{5.8}$$

Here $f$ is the number of "mass" parameters. Now the number of mass parameters is given by the rank of Mordell-Weil lattice (see formula. 5.5):

$$f = r_{MW} = \rho - 2 - (n_\infty - 1). \tag{5.9}$$

Finally, the number of bulk $A_1$ singularities and the topological invariant $\delta_\infty$ (assuming $h_\infty = 0$) satisfy the equation (see formula. 5.4) [10]:

$$\frac{g + 1}{2g + 1}(\#A_1 + \delta_\infty) = \rho - 2. \tag{5.10}$$

Combine the above three equations to eliminate $\#A_1$ and $f$, we have the following equation

$$(\rho - 2) - (n_\infty - 1) + 2g = (\rho - 2)\frac{2g + 1}{g + 1} - \delta_\infty,$$

---

[9]Notice that in general only a subset of these BPS particles are needed to generate the full charge lattice.

[10]In the general case, the left hand side of this equation should include a global contribution $\sigma_f$ of the fiberation. If $\rho = 4g + 6$ and the fiberation is hyperelliptic, one get a total of $8g + 4$ $A_1$ singularities.

and so we have
$$\delta_\infty - n_\infty = (\rho - 2)\frac{g}{g+1} - 2g - 1.$$
Take $\rho(S) = 4g + 6$ for the hyperelliptic fiberation, we have
$$\delta_\infty - n_\infty = 2g - 1.$$

In the general case, $f$ is not the rank of physical flavor symmetry, but is given by rank of MW group, see formula 5.5. Then one has the same constraint for the fiber at $\infty$. The above result then implies that the dual graph for $F_\infty$ has to be a **tree** of **rational** curves!

Motivated by the above results for hyperelliptic fiberation with maximal Picard number, we conjecture that the singular fiber at $\infty$ for 4d theory has to satisfy the condition 5.7.

*Comments*: We have used the correspondence of dual graph and 3d mirror to constrain the possible IR theory. In the UV case, this constraint might still be imposed on the dual fiber for $F_\infty$! The dual fiber is defined for periodic map, and one can define it for general fiber by taking the dual of the periodic part of the map.

**SCFT**: Let's consider 4d SCFT which is given by a periodic map with singular fiber $F$. There is a natural candidate for the fiber at $\infty$: the dual periodic map $F'$. We first consider a fibered surface with just two singular fibers $F$ and $F'$, and the topological constraints are automatically satisfied, see equation. 5.2. Then one can deform the fiber $F$ into $A_1$ singularities to get the global SW geometry.

*Example*: Let's consider $(A_1, A_{n-1})$ theory with $n$ even, and the corresponding periodic map for fiber $F$ is $(n, g' = 0, \frac{1}{n} + \frac{1}{n} + \frac{n-2}{n})$, $g = \frac{n-2}{2}$ (the Euler number is $\delta_x = n - 1$), the dual fiber is given by the data $(n, g' = 0, \frac{n-1}{n} + \frac{n-1}{n} + \frac{2}{n})$ (the Euler number is $\delta_x = 3n - 3$). The total Euler number satisfies the condition $\delta_x(F) + \delta_x(F') = 4n - 4 = 8g + 4$. This implies that: a): The global curve can be given by the fiberation of hyperelliptic curves; b): $d_x = \delta_x$ for $F$ and $F'$; c): The generic deformation of the theory has only $A_1$ singularities, which agrees with the result using the singularity theory (one can engineer the SCFT by an isolated hypersurface singularity [10], and generic deformation of singularity has only $A_1$ singularities).

One can similarly find the singular fiber at $\infty$ for other SCFTs discussed in section. 4.3: one simply choose the dual periodic map in the decomposition of $F$. See figure. 29 for an example.

*Global curve*: The global curves for various SCFTs have been found for many class of theories, and they are useful to verity our general strategy of classifying theories. Here we give the global curve for $D_2(SU(n+1))$ theory: when $n$ is odd, it is just $SU(\frac{n+1}{2})$ coupled with $n + 1$ fundamental flavor; When $n$ is even, it is an interacting AD theory with flavor symmetry group $SU(n+1)$ [7, 42]. The global curve takes the following simple form
$$y^2 = x^{n+1} + p_2 x^{n-1} + \ldots + p_n x + p_{n+1} + t^2.$$

Here $t$ is the Coulomb branch operator (the scaling dimension is $[t] = \frac{n+1}{2}$). The Mordell-Weil lattice is $A_n$ ($A_n \oplus U(1)$) for $n$ even ($n$ odd) [38], which is in consistent with the flavor symmetry of the physical theory.

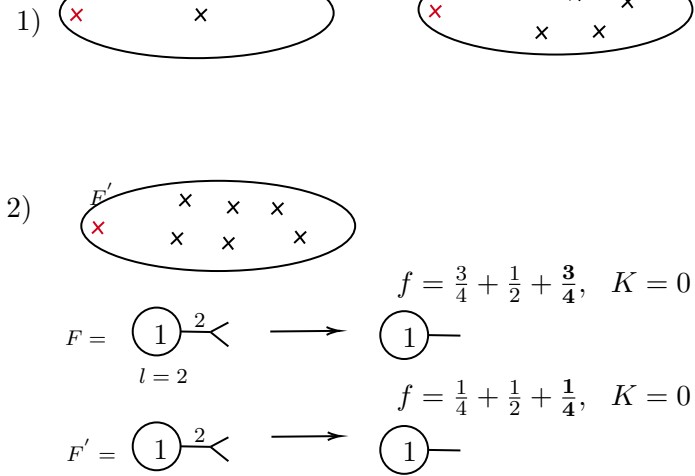

**Figure 29**: (1): Left: The global geometry of a SCFT defined by periodic map $F$, and $F'$ is the dual map; They can sit together to get an isotrivial family; Right: $F$ can be deformed into several $A_1$ singularities which give rise to global SW geometry for the theory $F$. (2): A SCFT is represented by a fiber $F$, and the fiber at infinity is given by its dual $F'$.

**Asymptotical free theory**: Let's now give some global SW geometries for asymptotical free theory.

$SU(n)$ *gauge theory coupled with $n_f$ flavor*: The singular fiber at $\infty$ is given by the data shown in figure. 30. This gives the SW geometry for $SU(n)$ gauge theory coupled with $n_f = 2n - K$, $K \geq 1$ hypermultiplets in fundamental representation. A simple check is to compute the topological Euler number $\delta_x$ (which is conjectured to be the same as the holomorphic invariant $d_x$), and the answer is

$$d_x = \delta_x = 2n + K + 2(n-1) = 4n + K - 2.$$

So the number of $A_1$ singularities at the bulk are given as (since the global SW curve is hyperelliptic):

$$8(n-1) + 4 - (4n + K - 2) = 2(n-1) + 2n - K = 2r + f.$$

The singular fiber for $Sp(2n)$ gauge theory coupled with $n_f = 2n + 2 - K$ hypermultiplets in fundamental representation is also shown in figure. 30.

*Pure $SU(n)$ gauge theory*: Let's consider pure $SU(n)$ gauge theory in more detail, and there are a total of $2n - 2$ $A_1$ singularities at the bulk. We'd like to determine other special vacua by simple topological constraints: a) The Euler number is less than $2n - 2$; b) There is no flavor symmetry so the dual graph has just one node. Here are some obvious possibilities:

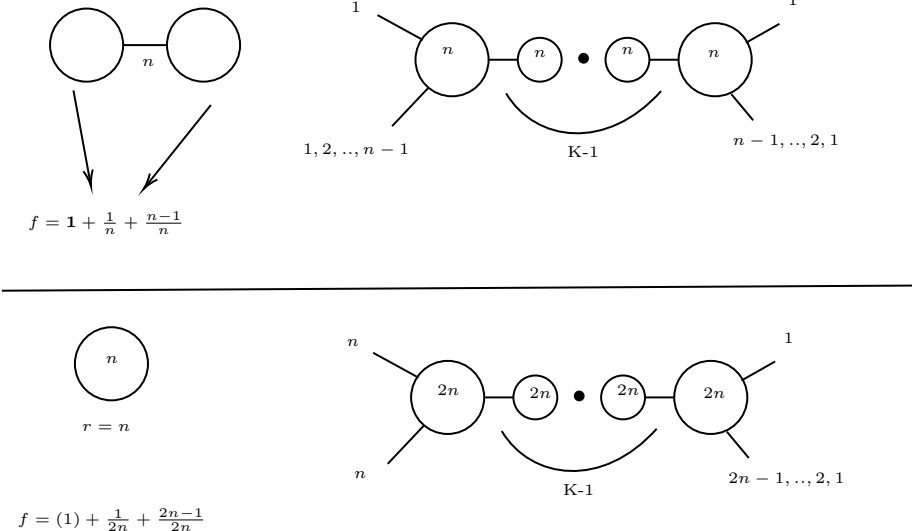

**Figure 30**: Upper: The singular fiber at $\infty$ for $SU(n)$ gauge theory with $2n - K$ flavors; The weighted graph is given by two genus zero curve connected by an edge with multiplicity $n$. The periodic map on each genus zero component is $\frac{1}{n} + \frac{n-1}{n} + \mathbf{1}$. Bottom: The singular fiber at $\infty$ for $Sp(2n)$ gauge theory with $2n + 2 - K$ flavor; The weighted graph is given by a single component with $r = n$ cutting curves, which are amphidrome. The periodic map on the irreducible components are $(1) + \frac{1}{2n} + \frac{n-1}{2n}$.

1. The degeneration is given by the weighted graph $(g = n - 1, r = g)$, namely there are $n - 1$ non-separating cutting curves, see figure. 31. The periodic map on the component $\Sigma/C$ is trivial. To have no flavor symmetry, the linking number $K = 1$, so the IR theory is direct sum of $U(1)$ with one hypermultiplet: $(U(1) \oplus 1)^{n-1}$. The Euler number of this vacua is $n - 1$. This particular vacua is important as it gives the confining vacua for the corresponding $\mathcal{N} = 1$ theory.

2. The degeneration gives AD theory and free vector multiplets. Depending $n$, the AD theory might have a $U(1)$ global symmetry which would be gauged.

$\mathcal{N} = 2^*$ $SU(n)$ *gauge theory*: Although we mainly focus on the case where the bulk singularities are just $A_1$ singularities. It is actually possible to get the SW solution for other theories by considering more general un-deformable singularities. Notice that the local structure of the singularities are still classified by the result of this paper: one just do not deform those singularities into $A_1$ type. We leave the general discussion to a separate publication, and here give the results for $\mathcal{N} = 2^*$ $SU(n)$ gauge theory. The fiber at $\infty$ should give a periodic map, which gives the scaling dimensions $2, 3, \ldots, n$ [3]. The natural candidate is the one for $SU(n)$ with $n_f = 2n$, namely, it is given by the period map with data $(n, g' = 0, \frac{1}{n} + \frac{1}{n} + \frac{n-1}{n} + \frac{n-1}{n})$. For the bulk singularities, there are following two constraints a): The topological constraints, namely, $\sum_{bulk} \delta_x = 4n - 2$; b): Dirac quantization [12]; c): The rank of charge lattice is $2n - 1$ and so there would be at least

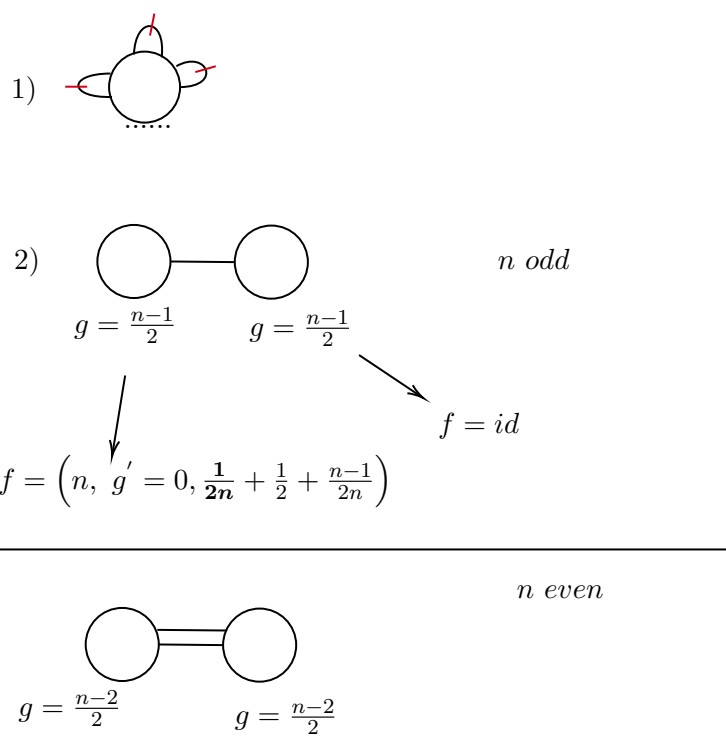

1)

2) $\qquad$ $n \ odd$

$g = \frac{n-1}{2}$ $\qquad$ $g = \frac{n-1}{2}$

$f = id$

$f = \left(n, \ g' = 0, \mathbf{\frac{1}{2n}} + \frac{1}{2} + \frac{n-1}{2n}\right)$

$n \ even$

$g = \frac{n-2}{2}$ $\qquad$ $g = \frac{n-2}{2}$

$f = \left(n, \ g' = 0, \mathbf{\frac{1}{n}} + \mathbf{\frac{1}{n}} + \frac{n-2}{n}\right)$ $\qquad$ $f = id$

**Figure 31**: (1): The special vacua for pure $SU(n)$ theory, and there are $g$ non-separating curves in the cut system and all the associated integers $K = 1$. The IR theory is just g copies of $U(1)$ coupled with one hypermultiplet; (2) The singular fiber corresponding to AD theory plus free vector multiplets.

$2n - 1$ singularities. It would be interesting to find out the type of these local singularities.

$\mathcal{N} = 2^*$ $Sp(2n)$ *gauge theory*: The fiber at $\infty$ is given by periodic map with data $(n, g' = 0, \frac{1}{2n} + \frac{1}{2} + \frac{1}{2} + \frac{2n-1}{2n})$, and its topological Euler number is $4n + 2$. It would be interesting to find out the type of bulk singularities.

*Linear quiver*: Let's consider the UV complete linear quiver $n_1 - SU(n) - SU(n) - n_2$, here $n_1, n_2 < n$. It is natural to state that the degeneration at $\infty$ should be the one shown in figure. 32, the data is $n_1 = n - K_1, \ \ n_2 = n - K_2$. Since the SW geometry is no longer hyperelliptic and the Picard number is not known, and so one do not know the number of $A_1$ singularities at the bulk. We notice that there is a SCFT by taking $n_1 = n_2 = n$, and the total Euler number on the global SW geometry can be easily computed: the corresponding periodic map is self-dual (with valency data $(\frac{1}{n})^3 + (\frac{n-1}{n})^3$, and $\delta_x = 7n - 4$), so the total Euler number is $e_{total} = 14n - 8$. We assume that the total Euler number should be still the same for the general $n_1, n_2$ case, and this gives rise to the number of bulk $A_1$ singularities

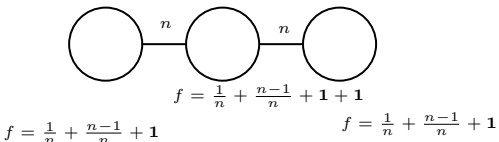

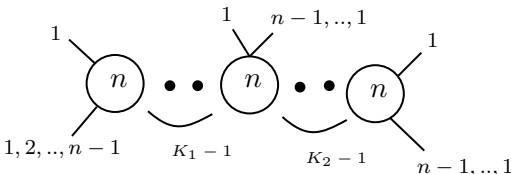

**Figure 32**: Up: the pseudo-periodic map for a UV complete linear quiver theory; Bottom: the dual graph and the topological Euler number is $\delta_x = 7n + K_1 + K_2 - 4$.

as

$$\#A_1 = 14n - 8 - \delta_\infty = 14n - 8 - (7n + 2n - n_1 - n_2 - 4)$$
$$= 5n + n_1 + n_2 - 4 = 2r + f + (n - 1).$$

Here $r = 2n - 2$ is the rank of the theory, $f = n_1 + n_2 + 1$ are the number of mass parameters.

## 5.2  5d theory

Let's now turn to the SW geometry for 5d KK theory: the Coulomb branch solution for 5d $\mathcal{N} = 1$ theory compactified on a circle with finite size. The difference with 4d SW geometry is that the charge lattice has dimension $\Gamma = 2r + f + 1$, where the extra one denotes the KK charge. Following similar argument as 4d theory, the topological constraint for a singular fiber at $\infty$ of a 5d theory is:

$$\delta_\infty - n_\infty = 2g - 2.$$

Hence the dual graph of $F$ satisfies following condition: a) All the components are all rational curves; b) It has just one loop; c): The dual fiber is good in the sense that one can get a good 3d mirror.

*Example*: Let's consider several class of possible UV singular fibers for 5d rank 3 KK theory, see figure. 33. Some of them might be related to 5d theory engineered using toric diagram (such that there is a $SU(4)$ gauge theory interpretation along certain special locus.) [43]. A check is that one can compute the eigenvalues of monodromy around infinity using the method explained in [4], and there are two eigenvalues 1, and other eigenvalues are roots of order 3; on the other hand, for the pseudo-periodic map shown in figure. 33, the induced monodromy on homology cycles also has two eigenvalues 1, and other eigenvalues are of order 3 (the order of the periodic map is 3.).

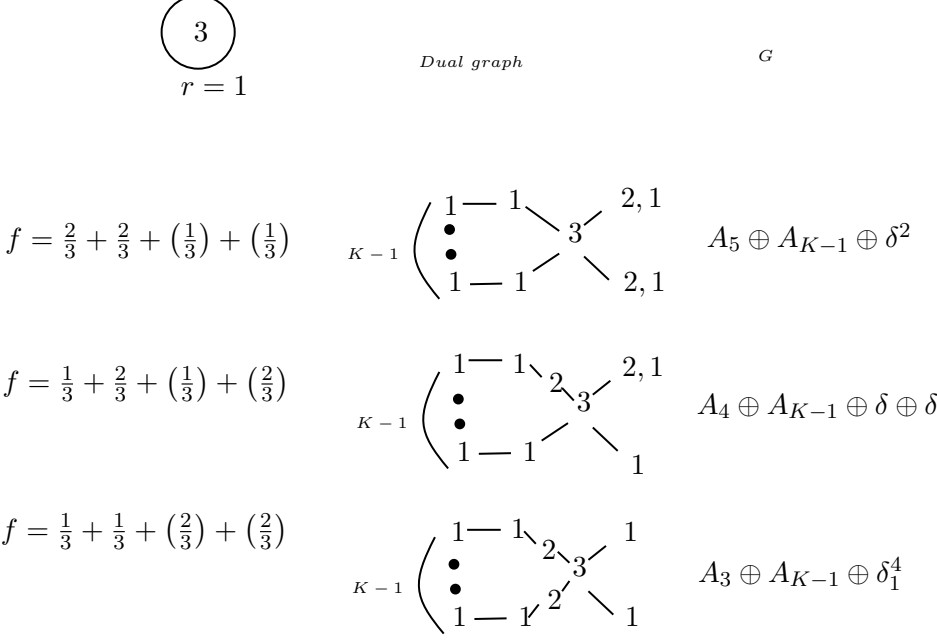

**Figure 33**: Singular fiber which might give rise to Coulomb branch geometry for 5d $\mathcal{N} = 1$ KK theories. Here $\delta$ has self-intersection number $-4$, and $\delta_1$ has self-intersection number $-3$.

### 5.3 6d theory

Let's now turn to the SW geometry for 6d KK theory: the Coulomb branch solution for 6d $(1,0)$ theory compactified on a torus with finite size. The difference with 4d SW geometry is that the charge lattice has dimension $\Gamma = 2r + f + 2$, where the extra two denotes the KK charge along two cycles of the torus. Following similar argument as 4d theory, the topological constraint for a 4d theory is

$$\delta_\infty - n_\infty = 2g - 3.$$

Hence the dual graph of $F$ satisfies following condition: a) All the components are all rational curves; b) It has just two loops; c): The dual fiber is good in the sense that one can get a good 3d mirror for it.

*Example*: A class of possible UV singular fiber for 6d rank 3 KK theory are shown in figure. 34.

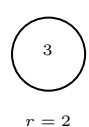

$$r = 2$$

$$f = \left(\tfrac{1}{2}\right) + \left(\tfrac{1}{2}\right) + \left(\tfrac{1}{2}\right) + \left(\tfrac{1}{2}\right)$$

$$G = A_1 \oplus A_{K_1 - 1} \oplus A_{K_1 - 1} \oplus \delta_1^4$$

**Figure 34**: Singular fiber that might give rise to the SW geometry for rank three 6d $(1,0)$ KK theories.

## 6 Conclusion

We give a classification of IR and UV behavior for Coulomb branch geometry of theories with eight supercharges, with the following assumptions: a) the local SW solution is given by the fiberation of genus $g$ curves; b): The theory is associated with the generic deformation of the singularity; c): A one parameter slice of SW geometry is taken. The crucial fact is that such degeneration is classified by the conjugacy class of mapping class group [17], which are the so-called pseudo-periodic map of negative type. The description of such conjugacy class is combinatorial, which makes a systematical study possible.

To determine the IR theory (associated with the singular fiber at bulk) or the UV theory (associated with the singular fiber at $\infty$), we find the dual graph from the pseudo-periodic map play a crucial role: they are closed related to 3d mirror for these theories. Using the dual graph and the related 3d mirror, we can identify SCFTs (such as Argyres-Douglas theories, $T_n$ theories), IR free theories, and UV complete theories with certain conjugacy class of mapping class group! Given the simplicity of the description, we believe that many physical questions about theories with eight supercharges can be solved! In this paper, it helps us explain the number of $A_1$ singularities in the generic deformation of $T_n$ theories, and confirms the 3d mirror proposal for Argyres-Douglas theories [23].

There are several further interesting questions that one can study:

1. The usual electric-magnetic duality group acts on photon coupling, and it turns out that this action along is not enough to determine the IR theory. The duality group is interpreted geometrically as the action on homology groups. Now if one look at enlarged action and consider the conjugacy class in mapping class group, then the IR theory can be completely determined. The question is the physical objects that the mapping class group acts on. Since mapping class group naturally acts on the simple closed curves on Riemann surface, which is in turn gives the IR line operators, this

suggests that one need to study the action on the set of IR line operators when one go along the special vacua!

2. It would be interesting to give an interpretation for the mysterious relation between the dual graph and the 3d mirror. We have used the identification of 3d mirror to constrain the possible appearance of singular fiber. It would be interesting to study the physical meaning of these constraints.

3. We have studied one parameter slice of the full Coulomb branch geometry, and it is certainly useful to study the full parameter space of SW geometry, for instance, one can not find a one parameter scale-invariant geometry for a SCFT, but it is possible to find a scale-invariant geometry for a multiple parameter family [44]. More generally, one might find more constraints on the appearance of singular fibers by looking at the full deformation space of the physical theory.

4. Certainly, the next major question is the classification of global SW geometry (see [45] for rank one case). We hope to report the progress on this question in the near future.

5. In our study, the IR theory is assumed to be associated with generic deformation. To give a complete classification, one need to classify the theory associated with the so-called non-generic deformation.

6. We also assume the SW geometry is given by a family of Riemann surfaces. This condition can be relaxed in following way: one can consider a finite group action on the local family, and study the quotient. In particular, it is possible to define a so-called "orientifold" action on the conjugacy class to get quiver theories with orthosymplectic gauge groups. Details will appear in a separate publication.

### Acklowledgement

DX would like to thank Prof. S. Y. Cheng for constant encouragement.

# A  Periodic maps for $1 \leq g \leq 5$

Non-identical conjugacy classes of periodic maps of closed surfaces of genus $1 \leq g \leq 5$ are listed in table. 2,3,4,5,6.

| Order | Data |
|---|---|
| $n = 6$ | $(\frac{1}{6} + \frac{1}{3} + \frac{1}{2}, \frac{5}{6} + \frac{2}{3} + \frac{1}{2})$ |
| $n = 4$ | $(\frac{1}{4} + \frac{1}{4} + \frac{1}{2}, \frac{3}{4} + \frac{3}{4} + \frac{1}{2})$ |
| $n = 3$ | $(\frac{1}{3} + \frac{1}{3} + \frac{1}{3}, \frac{2}{3} + \frac{2}{3} + \frac{2}{3})$ |
| $n = 2$ | $\frac{1}{2} + \frac{1}{2} + \frac{1}{2} + \frac{1}{2}$ |

**Table 2**: The periodic maps for genus $g = 1$. The genus $g^{'} = 0$ is ignored.

| Order | Data |
|---|---|
| $n = 10$ | $(\frac{1}{10} + \frac{2}{5} + \frac{1}{2}, \frac{9}{10} + \frac{3}{5} + \frac{1}{2})$, $(\frac{3}{10} + \frac{1}{5} + \frac{1}{2}, \frac{7}{10} + \frac{4}{5} + \frac{1}{2})$, |
| $n = 8$ | $(\frac{1}{8} + \frac{3}{8} + \frac{1}{2}, \frac{7}{8} + \frac{5}{8} + \frac{1}{2})$ |
| $n = 6$ | $(\frac{1}{6} + \frac{1}{6} + \frac{2}{3}, \frac{5}{6} + \frac{5}{6} + \frac{1}{3})$,  $(\frac{1}{3} + \frac{2}{3} + \frac{1}{2} + \frac{1}{2})$ |
| $n = 5$ | $(\frac{1}{5} + \frac{1}{5} + \frac{3}{5}, \frac{4}{5} + \frac{4}{5} + \frac{2}{5})$,  $(\frac{1}{5} + \frac{1}{5} + \frac{2}{5}, \frac{4}{5} + \frac{4}{5} + \frac{3}{5})$ |
| $n = 4$ | $\frac{1}{4} + \frac{3}{4} + \frac{1}{2} + \frac{1}{2}$ |
| $n = 3$ | $\frac{1}{3} + \frac{1}{3} + \frac{2}{3} + \frac{2}{3}$ |
| $n = 2$ | $\frac{1}{2} + \frac{1}{2} + \frac{1}{2} + \frac{1}{2} + \frac{1}{2} + \frac{1}{2}$ |
| $g^{'} = 1, n = 2$ | $\frac{1}{2} + \frac{1}{2}$ |

**Table 3**: The periodic maps for genus $g = 2$.

| Order | Data |
|---|---|
| $n = 14$ | $(\frac{3}{14} + \frac{2}{7} + \frac{1}{2}, \frac{11}{14} + \frac{5}{7} + \frac{1}{2})$, $(\frac{1}{14} + \frac{3}{7} + \frac{1}{2}, \frac{13}{14} + \frac{4}{7} + \frac{1}{2})$ $(\frac{5}{14} + \frac{1}{7} + \frac{1}{2}, \frac{9}{14} + \frac{6}{7} + \frac{1}{2})$ |
| $n = 12$ | $(\frac{1}{12} + \frac{5}{12} + \frac{1}{2}, \frac{11}{12} + \frac{7}{12} + \frac{1}{2})$, $(\frac{1}{12} + \frac{1}{4} + \frac{2}{3}, \frac{11}{12} + \frac{3}{4} + \frac{1}{3})$ $(\frac{5}{12} + \frac{1}{4} + \frac{1}{3}, \frac{7}{12} + \frac{3}{4} + \frac{2}{3})$ |
| $n = 9$ | $(\frac{1}{9} + \frac{5}{9} + \frac{1}{3}, \frac{8}{9} + \frac{5}{7} + \frac{1}{2})$, $(\frac{1}{9} + \frac{2}{9} + \frac{2}{3}, \frac{8}{9} + \frac{7}{9} + \frac{1}{3})$ $(\frac{2}{9} + \frac{4}{9} + \frac{1}{3}, \frac{7}{9} + \frac{5}{9} + \frac{2}{3})$ |
| $n = 8$ | $(\frac{1}{8} + \frac{5}{8} + \frac{1}{4}, \frac{7}{8} + \frac{3}{8} + \frac{3}{4})$, $(\frac{1}{8} + \frac{1}{8} + \frac{3}{4}, \frac{7}{8} + \frac{7}{8} + \frac{1}{4})$ $(\frac{3}{8} + \frac{3}{8} + \frac{1}{4}, \frac{5}{8} + \frac{5}{8} + \frac{3}{4})$ |
| $n = 7$ | $(\frac{1}{7} + \frac{1}{7} + \frac{5}{7}, \frac{6}{7} + \frac{6}{7} + \frac{2}{7})$, $(\frac{1}{7} + \frac{2}{7} + \frac{4}{7}, \frac{6}{7} + \frac{5}{7} + \frac{3}{7})$ $(\frac{1}{7} + \frac{3}{7} + \frac{3}{7}, \frac{6}{7} + \frac{4}{7} + \frac{4}{7})$ |
|  | $(\frac{2}{7} + \frac{2}{7} + \frac{3}{7}, \frac{5}{7} + \frac{5}{7} + \frac{4}{7})$ |
| $n = 6$ | $(\frac{5}{6} + \frac{1}{6} + \frac{1}{2} + \frac{1}{2}), (\frac{5}{6} + \frac{1}{3} + \frac{1}{3} + \frac{1}{2}, \frac{1}{6} + \frac{2}{3} + \frac{2}{3} + \frac{1}{2})$ |
| $n = 4$ | $(\frac{1}{4} + \frac{1}{4} + \frac{1}{4} + \frac{1}{4}, \frac{3}{4} + \frac{3}{4} + \frac{3}{4} + \frac{3}{4})$, $(\frac{1}{4} + \frac{1}{4} + \frac{1}{2} + \frac{1}{2} + \frac{1}{2}, \frac{3}{4} + \frac{3}{4} + \frac{1}{2} + \frac{1}{2} + \frac{1}{2}), (\frac{3}{4} + \frac{3}{4} + \frac{1}{2} + \frac{1}{2})$ |
| $n = 3$ | $(\frac{2}{3} + \frac{2}{3} + \frac{2}{3} + \frac{2}{3} + \frac{1}{3}, \frac{1}{3} + \frac{1}{3} + \frac{1}{3} + \frac{1}{3} + \frac{2}{3})$ |
| $n = 2$ | $\frac{1}{2} + \frac{1}{2} + \frac{1}{2} + \frac{1}{2} + \frac{1}{2} + \frac{1}{2} + \frac{1}{2} + \frac{1}{2}$ |
| $g' = 1, n = 4$ | $\frac{1}{2} + \frac{1}{2}$ |
| $g' = 1, n = 3$ | $\frac{1}{3} + \frac{2}{3}$ |
| $g' = 1, n = 2$ | $\frac{1}{2} + \frac{1}{2} + \frac{1}{2} + \frac{1}{2}$ |
| $g' = 2, n = 2$ | $f : \Pi \to \Pi'$ is an umramified covering. |

**Table 4**: The periodic maps for genus $g = 3$.

## B   Rank 4 and rank 5 4d SCFTs from periodic map

We list the SCFTs associated with the periodic maps for genus 4 case (figure. 35) and genus 5 case (figure. 36).

| Order | Data |
|---|---|
| $n = 18$ | $(\frac{1}{2} + \frac{1}{9} + \frac{7}{18}, \frac{1}{2} + \frac{8}{9} + \frac{11}{18})$, $\quad(\frac{1}{2} + \frac{2}{9} + \frac{5}{18}, \frac{1}{2} + \frac{7}{9} + \frac{13}{18})$ $\quad(\frac{1}{2} + \frac{4}{9} + \frac{1}{18}, \frac{1}{2} + \frac{5}{9} + \frac{17}{18})$ |
| $n = 16$ | $(\frac{1}{2} + \frac{3}{16} + \frac{5}{16}, \frac{1}{2} + \frac{13}{16} + \frac{11}{16})$, $\quad(\frac{1}{2} + \frac{7}{16} + \frac{1}{16}, \frac{1}{2} + \frac{9}{16} + \frac{15}{16})$ |
| $n = 15$ | $(\frac{1}{3} + \frac{1}{5} + \frac{7}{15}, \frac{2}{3} + \frac{4}{5} + \frac{8}{15})$, $\quad(\frac{1}{3} + \frac{2}{5} + \frac{4}{15}, \frac{2}{3} + \frac{3}{5} + \frac{11}{15})$ $\quad(\frac{1}{5} + \frac{2}{3} + \frac{2}{15}, \frac{4}{5} + \frac{1}{3} + \frac{13}{15})$ |
|  | $(\frac{3}{5} + \frac{1}{3} + \frac{1}{15}, \frac{2}{5} + \frac{2}{3} + \frac{14}{15})$ |
| $n = 10$ | $(\frac{1}{2} + \frac{1}{2} + \frac{2}{5} + \frac{3}{5})$, $(\frac{2}{5} + \frac{3}{10} + \frac{3}{10}, \frac{3}{5} + \frac{7}{10} + \frac{7}{10})$ $\quad(\frac{1}{10} + \frac{3}{10} + \frac{3}{5}, \frac{9}{10} + \frac{7}{10} + \frac{2}{5})$ |
|  | $(\frac{1}{10} + \frac{7}{10} + \frac{1}{5}, \frac{9}{10} + \frac{3}{10} + \frac{4}{5})$ $\quad(\frac{1}{10} + \frac{1}{10} + \frac{4}{5}, \frac{9}{10} + \frac{9}{10} + \frac{1}{5})$ |
| $n = 9$ | $(\frac{1}{9} + \frac{4}{9} + \frac{4}{9}, \frac{8}{9} + \frac{5}{9} + \frac{5}{9})$, $\quad(\frac{1}{9} + \frac{1}{9} + \frac{7}{9}, \frac{8}{9} + \frac{8}{9} + \frac{2}{9})$ $\quad(\frac{2}{9} + \frac{2}{9} + \frac{5}{9}, \frac{7}{9} + \frac{7}{9} + \frac{4}{9})$ |
| $n = 8$ | $(\frac{1}{2} + \frac{1}{2} + \frac{1}{8} + \frac{7}{8})$, $(\frac{1}{2} + \frac{1}{2} + \frac{3}{8} + \frac{5}{8})$ |
| $n = 6$ | $(\frac{1}{2} + \frac{1}{2} + \frac{1}{2} + \frac{1}{3} + \frac{1}{6}, \frac{1}{2} + \frac{1}{2} + \frac{1}{2} + \frac{2}{3} + \frac{5}{6})$, $(\frac{1}{6} + \frac{1}{6} + \frac{1}{3} + \frac{1}{3}, \frac{5}{6} + \frac{5}{6} + \frac{2}{3} + \frac{2}{3})$, $\quad(\frac{5}{6} + \frac{1}{6} + \frac{2}{3} + \frac{1}{3})$ |
|  | $(\frac{1}{6} + \frac{1}{6} + \frac{1}{6} + \frac{1}{2}, \frac{5}{6} + \frac{5}{6} + \frac{5}{6} + \frac{1}{2})$, $(\frac{1}{3} + \frac{1}{3} + \frac{1}{3} + \frac{1}{2} + \frac{1}{2}, \frac{2}{3} + \frac{2}{3} + \frac{2}{3} + \frac{1}{2} + \frac{1}{2})$ |
| $n = 5$ | $(\frac{1}{5} + \frac{1}{5} + \frac{1}{5} + \frac{2}{5}, \frac{4}{5} + \frac{4}{5} + \frac{4}{5} + \frac{1}{5})$, $\quad(\frac{2}{5} + \frac{2}{5} + \frac{2}{5} + \frac{4}{5}, \frac{3}{5} + \frac{3}{5} + \frac{3}{5} + \frac{1}{5})$ |
|  | $(\frac{2}{5} + \frac{2}{5} + \frac{3}{5} + \frac{3}{5}))$, $(\frac{4}{5} + \frac{4}{5} + \frac{1}{5} + \frac{1}{5})$ |
| $n = 4$ | $(\frac{1}{4} + \frac{1}{2} + \frac{1}{2} + \frac{1}{2} + \frac{1}{2} + \frac{3}{4})$, $\quad(\frac{1}{4} + \frac{1}{4} + \frac{1}{4} + \frac{1}{2} + \frac{3}{4}, \frac{3}{4} + \frac{3}{4} + \frac{3}{4} + \frac{1}{2} + \frac{1}{4})$ |
| $n = 3$ | $(\frac{1}{3} + \frac{1}{3} + \frac{1}{3} + \frac{1}{3} + \frac{1}{3} + \frac{1}{3}, \frac{2}{3} + \frac{2}{3} + \frac{2}{3} + \frac{2}{3} + \frac{2}{3} + \frac{2}{3})$, $(\frac{2}{3} + \frac{2}{3} + \frac{2}{3} + \frac{1}{3} + \frac{1}{3} + \frac{1}{3})$ |
| $n = 2$ | $\frac{1}{2} + \frac{1}{2} + \frac{1}{2} + \frac{1}{2} + \frac{1}{2} + \frac{1}{2} + \frac{1}{2} + \frac{1}{2} + \frac{1}{2} + \frac{1}{2}$ |
| $g' = 1, n = 6$ | $\frac{1}{2} + \frac{1}{2}$ |
| $g' = 1, n = 4$ | $\frac{1}{4} + \frac{3}{4}$ |
| $g' = 1, n = 3$ | $(\frac{1}{3} + \frac{1}{3} + \frac{1}{3}, \frac{2}{3} + \frac{2}{3} + \frac{2}{3})$ |
| $g' = 1, n = 2$ | $\frac{1}{2} + \frac{1}{2} + \frac{1}{2} + \frac{1}{2} + \frac{1}{2} + \frac{1}{2}$ |
| $g' = 2, n = 3$ | Unramified covering |
| $g' = 2, n = 2$ | $\frac{1}{2} + \frac{1}{2}$ |

**Table 5**: The periodic maps for genus $g = 4$.

| Order | Data |
|-------|------|
| $n = 22$ | $(\frac{1}{2} + \frac{1}{11} + \frac{9}{22}, \frac{1}{2} + \frac{10}{11} + \frac{13}{22})$, $(\frac{1}{2} + \frac{2}{11} + \frac{7}{22}, \frac{1}{2} + \frac{9}{11} + \frac{15}{22})$ $(\frac{1}{2} + \frac{3}{11} + \frac{5}{22}, \frac{1}{2} + \frac{8}{11} + \frac{17}{22})$ $(\frac{1}{2} + \frac{4}{11} + \frac{3}{22}, \frac{1}{2} + \frac{7}{11} + \frac{19}{22})$, $(\frac{1}{2} + \frac{5}{11} + \frac{1}{22}, \frac{1}{2} + \frac{6}{11} + \frac{21}{22})$ |
| $n = 20$ | $(\frac{1}{2} + \frac{1}{20} + \frac{9}{20}, \frac{1}{2} + \frac{11}{20} + \frac{19}{20})$, $(\frac{1}{2} + \frac{3}{20} + \frac{7}{20}, \frac{1}{2} + \frac{17}{20} + \frac{13}{20})$ |
| $n = 15$ | $(\frac{1}{3} + \frac{2}{15} + \frac{8}{15}, \frac{2}{3} + \frac{13}{15} + \frac{7}{15})$, $(\frac{2}{3} + \frac{1}{15} + \frac{4}{15}, \frac{1}{3} + \frac{14}{15} + \frac{11}{15})$ |
| $n = 12$ | $(\frac{1}{6} + \frac{5}{12} + \frac{5}{12}, \frac{5}{6} + \frac{7}{12} + \frac{7}{12})$, $(\frac{1}{12} + \frac{1}{12} + \frac{5}{6}, \frac{11}{12} + \frac{11}{12} + \frac{1}{6})$ |
| $n = 11$ | $(\frac{1}{11} + \frac{1}{11} + \frac{9}{11}, \frac{10}{11} + \frac{10}{11} + \frac{2}{11})$, $(\frac{1}{11} + \frac{2}{11} + \frac{8}{11}, \frac{10}{11} + \frac{9}{11} + \frac{3}{11})$ $(\frac{1}{11} + \frac{3}{11} + \frac{7}{11}, \frac{10}{11} + \frac{8}{11} + \frac{4}{11})$ $(\frac{1}{11} + \frac{4}{11} + \frac{6}{11}, \frac{10}{11} + \frac{7}{11} + \frac{5}{11})$, $(\frac{1}{11} + \frac{5}{11} + \frac{5}{11}, \frac{10}{11} + \frac{6}{11} + \frac{6}{11})$ $(\frac{2}{11} + \frac{2}{11} + \frac{7}{11}, \frac{9}{11} + \frac{9}{11} + \frac{4}{11})$ $(\frac{2}{11} + \frac{4}{11} + \frac{5}{11}, \frac{9}{11} + \frac{7}{11} + \frac{6}{11})$, $(\frac{3}{11} + \frac{3}{11} + \frac{5}{11}, \frac{8}{11} + \frac{8}{11} + \frac{6}{11})$ $(\frac{2}{11} + \frac{3}{11} + \frac{6}{11}, \frac{9}{11} + \frac{8}{11} + \frac{5}{11})$ $(\frac{3}{11} + \frac{4}{11} + \frac{4}{11}, \frac{8}{11} + \frac{7}{11} + \frac{7}{11})$ |
| $n = 10$ | $(\frac{1}{2} + \frac{1}{2} + \frac{1}{10} + \frac{9}{10}), (\frac{1}{2} + \frac{1}{2} + \frac{3}{10} + \frac{7}{10})$ |
| $n = 8$ | $(\frac{1}{8} + \frac{1}{8} + \frac{1}{2} + \frac{1}{4}, \frac{7}{8} + \frac{7}{8} + \frac{1}{2} + \frac{3}{4})$, $(\frac{1}{2} + \frac{5}{8} + \frac{1}{8} + \frac{3}{4}, \frac{1}{2} + \frac{3}{8} + \frac{7}{8} + \frac{1}{4})$ $(\frac{1}{2} + \frac{3}{8} + \frac{3}{8} + \frac{3}{4}, \frac{1}{2} + \frac{5}{8} + \frac{5}{8} + \frac{1}{4})$ |
| $n = 6$ | $(\frac{1}{6} + \frac{1}{3} + \frac{1}{3} + \frac{2}{3} + \frac{1}{2}, \frac{5}{6} + \frac{2}{3} + \frac{2}{3} + \frac{1}{3} + \frac{1}{2})$, $(\frac{1}{6} + \frac{1}{6} + \frac{1}{2} + \frac{1}{2} + \frac{2}{3}, \frac{5}{6} + \frac{5}{6} + \frac{1}{2} + \frac{1}{2} + \frac{1}{3})$ $(\frac{1}{3} + \frac{2}{3} + \frac{1}{2} + \frac{1}{2} + \frac{1}{2} + \frac{1}{2})$, $(\frac{1}{6} + \frac{1}{6} + \frac{5}{6} + \frac{5}{6})$ |
| $n = 4$ | $(\frac{1}{4} + \frac{1}{4} + \frac{1}{4} + \frac{1}{4} + \frac{1}{2} + \frac{1}{2}, \frac{3}{4} + \frac{3}{4} + \frac{3}{4} + \frac{3}{4} + \frac{1}{2} + \frac{1}{2})$, $(\frac{1}{4} + \frac{1}{4} + \frac{1}{2} + \frac{1}{2} + \frac{3}{4} + \frac{3}{4})$ $(\frac{1}{4} + \frac{1}{4} + \frac{1}{2} + \frac{1}{2} + \frac{1}{2} + \frac{1}{2} + \frac{1}{2}, \frac{3}{4} + \frac{3}{4} + \frac{1}{2} + \frac{1}{2} + \frac{1}{2} + \frac{1}{2} + \frac{1}{2})$ |
| $n = 3$ | $(\frac{1}{3} + \frac{1}{3} + \frac{1}{3} + \frac{1}{3} + \frac{1}{3} + \frac{2}{3} + \frac{2}{3}, \frac{2}{3} + \frac{2}{3} + \frac{2}{3} + \frac{2}{3} + \frac{2}{3} + \frac{1}{3} + \frac{1}{3})$ |
| $n = 2$ | $(\frac{1}{2})^{12}$ |
| $g' = 1, n = 8$ | $\frac{1}{2} + \frac{1}{2}$ |
| $g' = 1, n = 6$ | $\frac{1}{3} + \frac{2}{3}$ |
| $g' = 1, n = 5$ | $\frac{1}{5} + \frac{4}{5}$ |
| $g' = 1, n = 4$ | $(\frac{1}{4} + \frac{1}{4} + \frac{1}{2}, \frac{3}{4} + \frac{3}{4} + \frac{1}{2})$, $(\frac{1}{2} + \frac{1}{2} + \frac{1}{2} + \frac{1}{2})$ |
| $g' = 1, n = 3$ | $\frac{1}{3} + \frac{2}{3} + \frac{1}{3} + \frac{2}{3}$ |
| $g' = 1, n = 2$ | $(\frac{1}{2})^8$ |
| $g' = 2, n = 4$ | Unramified covering |
| $g' = 2, n = 2$ | $(\frac{1}{2})^4$ |
| $g' = 3, n = 2$ | Unramified covering |

**Table 6**: The periodic maps for genus $g = 5$.

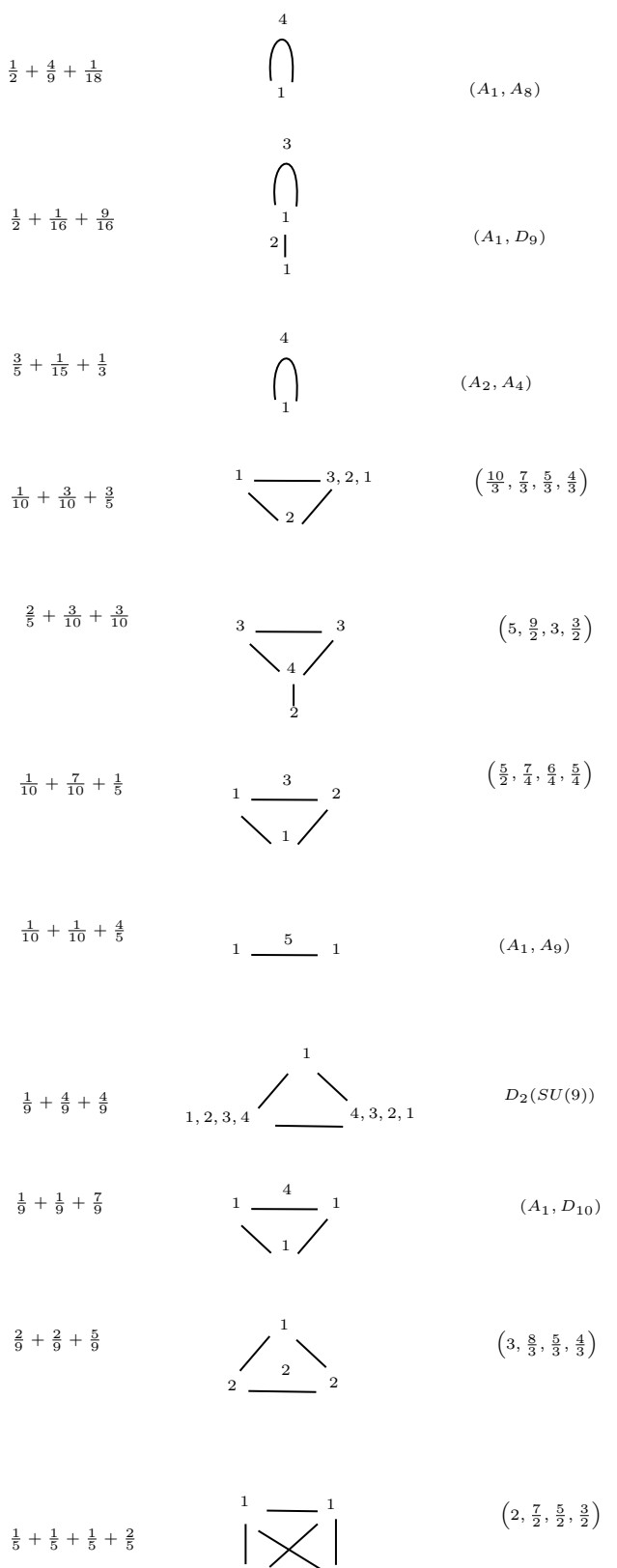

**Figure 35**: Rank 4 SCFTs and periodic map. Left: the data for periodic map; Middle: 3d mirror from dual graph; Right: the name for known theories or the scaling dimensions.

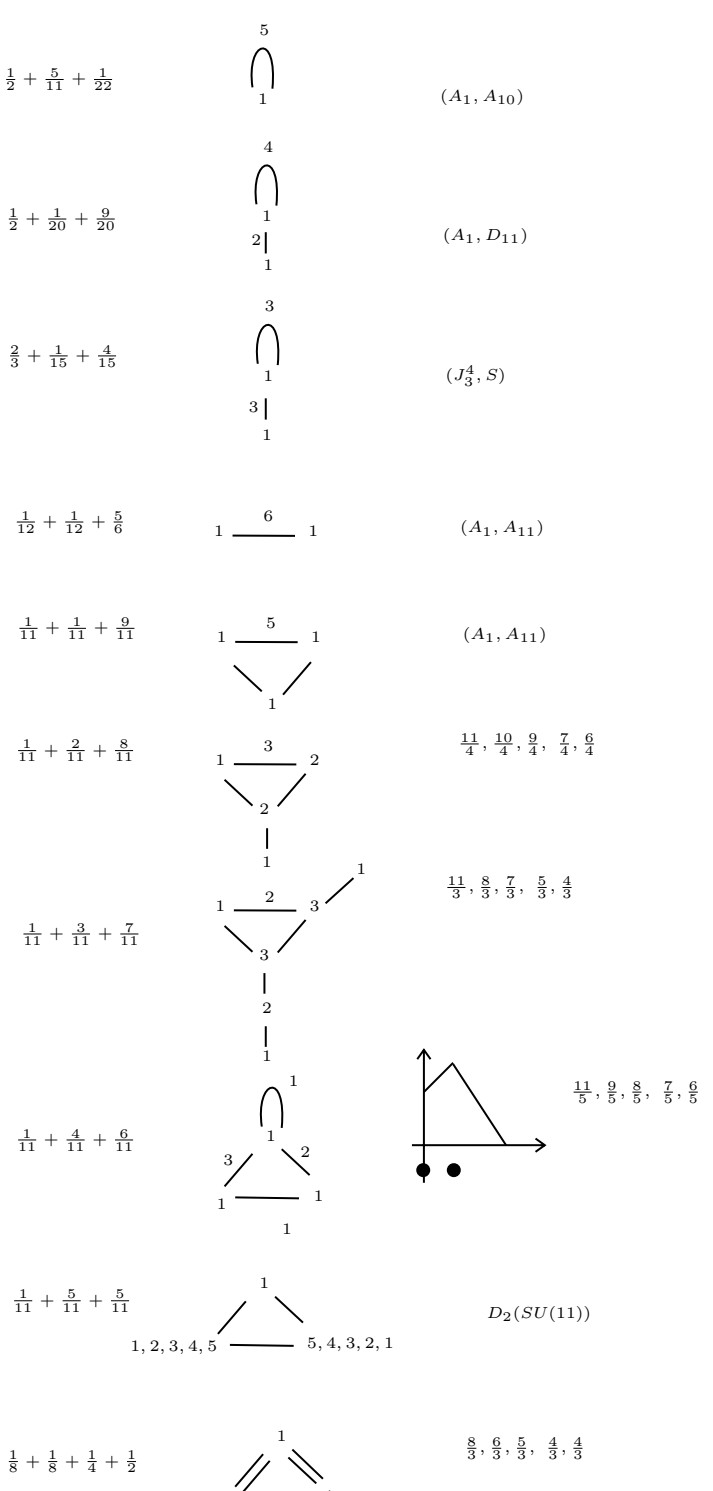

$\frac{1}{2} + \frac{5}{11} + \frac{1}{22}$ — $(A_1, A_{10})$

$\frac{1}{2} + \frac{1}{20} + \frac{9}{20}$ — $(A_1, D_{11})$

$\frac{2}{3} + \frac{1}{15} + \frac{4}{15}$ — $(J_3^4, S)$

$\frac{1}{12} + \frac{1}{12} + \frac{5}{6}$ — $(A_1, A_{11})$

$\frac{1}{11} + \frac{1}{11} + \frac{9}{11}$ — $(A_1, A_{11})$

$\frac{1}{11} + \frac{2}{11} + \frac{8}{11}$ — $\frac{11}{4}, \frac{10}{4}, \frac{9}{4}, \frac{7}{4}, \frac{6}{4}$

$\frac{1}{11} + \frac{3}{11} + \frac{7}{11}$ — $\frac{11}{3}, \frac{8}{3}, \frac{7}{3}, \frac{5}{3}, \frac{4}{3}$

$\frac{1}{11} + \frac{4}{11} + \frac{6}{11}$ — $\frac{11}{5}, \frac{9}{5}, \frac{8}{5}, \frac{7}{5}, \frac{6}{5}$

$\frac{1}{11} + \frac{5}{11} + \frac{5}{11}$ — $D_2(SU(11))$

$\frac{1}{8} + \frac{1}{8} + \frac{1}{4} + \frac{1}{2}$ — $\frac{8}{3}, \frac{6}{3}, \frac{5}{3}, \frac{4}{3}, \frac{4}{3}$

**Figure 36**: Rank 5 SCFTs and periodic maps. Left: the data for periodic map; Middle: 3d mirror from dual graph; Right: the name for known theories or the scaling dimensions.

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
