# Peer review of "Pseudo-periodic map and classification of theories with eight supercharges"

_SciPost Physics_

## Round 1 · Referee Report · Anonymous (Referee 1) · 2024-5-20

Strengths

  1. This is a skilful application of mathematical results to an important physical problem.
  2. The paper includes both a general discussion and a number of detailed examples.

Weaknesses

  1. Overall, the manuscript is full of grammatical mistakes and typos.
  2. Several points need to be explained more clearly.

Report

This is an interesting paper. The author studies theories with eight supercharges. A classification of the local behavior of Seiberg-Witten geometry around singularities of the Coulomb branch is provided. The author also gives some global Seiberg-Witten geometries.

I recommend the publication of this paper after the author addresses the following points and incorporate the changes requested.

  1. It is known that a given theory can have many different Seiberg-Witten curves, which lead to the same low energy effective theory and the same BPS spectrum, but contain different singularities. Some singularities do not correspond to the appearance of extra massless particles. It would be helpful to discuss how this subtlety would affect the analysis of this paper.

  2. Both "periodic map" and "pseudo-periodic map" are mentioned in the paper. It would be helpful to define them respectively and emphasize the difference between them.

  3. The classification of the Seiberg-Witten geometries is based on three assumptions. The assumption (b) assumes that only a one parameter slice of the Coulomb branch is considered, while the assumption (c) requires that the deformation is generic. How are these two assumptions consistent and sufficient?

  4. In line 3 of page 5, should it be periodic map or pseudo-periodic map?

  5. In line 11-12 of page 5, is the point $\infty$ on the physical Coulomb branch associated with the weakly coupled point? If yes, then what happens to theories without a weakly coupled point?

  6. In page 8, what is the difference between $\vec{C}_i$ and $C_i$?

  7. In page 10, the valency data is defined to be a group of numbers $(n\lambda_i, \lambda_i,\sigma_i)$. What is the meaning of "the formal sum of valency data" in the condition "Nielson theorem"?

  8. In Sec. 5.3, the Seiberg-Witten geometry for 6d KK theory is discussed by a straightforward generalization of the results in 4d and 5d. However, it is known that many well-defined 4d or 5d theories cannot be lifted to 6d due to anomaly. How does the anomaly cancellation condition affect the result?

  9. The author provides an alternative to the class S construction of the Seiberg-Witten geometry. To show the advantage of this new approach, the author should carefully discuss some Seiberg-Witten geometries (and the corresponding theories if possible) which are produced by the method in this paper but cannot be obtained from the class S construction.

Requested changes

  1. There are many grammatical mistakes and typos. The author should pay attention to the following issues: (1) a countable noun becomes plural by adding -s; (2) a countable noun always takes an article (a, an, the) when it is singular, and is followed by a verb modified by -s; (3) a subordinate clause cannot begin with a "here" (you can use "where"). It is highly desirable for the author to carefully read over the manuscript and fix the mistakes.

  2. In the last line of the paragraph "Screw number" in page 8, $S(C_i)$ should be $s(C_i)$.

  3. In line 4 of Sec. 3.1, $a_s$ should be replaced by $a_{l+1}$.

  4. Please make the format of Refs [18][26] compatible with the others.

Recommendation

Ask for minor revision

  • validity: high
  • significance: high
  • originality: high
  • clarity: good
  • formatting: good
  • grammar: below threshold

---

## Round 1 · Referee Report · Anonymous (Referee 2) · 2024-6-1

Strengths

  1. Exciting key idea that applies powerful mathematics to an important physics problem.

  2. Discussion of many examples.

Weaknesses

  1. Poor structure and grammar.

  2. Explanation of the physics is often weak.

  3. Fails to fully engage with the literature on the subject.

Report

This paper is built around a very interesting idea: let us try and classify rank-g 4d N=2 SQFTs by the local degeneration of the Seiberg-Witten curve (assumed to be a Riemann surface of genus g) on a dimension-1 subspace along the Coulomb branch (CB). This allows the author to use some powerful mathematics, due to Matsumoto and Montesinos-Amilibia (MM) [https://link.springer.com/book/10.1007/978-3-642-22534-5].

The key theorem of MM is that, roughly, local degenerations of one-parameter families of genus-g Riemann surfaces S are in 1-to-1 correspondence with conjugacy classes of the mapping class group of S. These can be represented by "pseudo-periodic maps" f on S, which can be written as some sorts of decorated graphs.

Thus it is proposed that classifying periodic maps can help classify both the IR and the UV of 4d N=2 theories. The author further identify the dual of that graph for f with the 3d mirror of the 4d N=2 IR theory that lives at the degeneration (at the price of some ad-hoc looking fiddling of the graph).

While the key idea is very interesting, the paper could have been much stronger. Beyond the bad grammar, the explanations given, both physical and mathematical, are often very poor. In fact, even for understanding the mathematical aspects of the construction, which the author spends a lot of time summarising, I would recommend looking at the book above instead.

As for the physics, one will not be readily convinced of the proposed classification after reading this paper. It would have been better to work out some examples in much more details, instead of only listing many examples without explaining all the reasoning and computation.

Below I list some questions and suggest some improvement that could help make the paper stronger.

The author also fails to engage with recent works by various other groups, focussing almost exclusively on their own previous work. This does not help the reader, even the one with expertise, to understand the proper contribution made here.

Despite these drawbacks, this is a rather stimulating paper and I would fully recommend it for publication after some minor revision to improve the presentation and the explanations.

Requested changes

  1. First recommendation is to thoroughly proofread and correct the English grammar and spelling.

Focussing on the physics here, the following simple questions naturally arise:

  1. Why can we focus on dimension-1 subspaces of the CB? The author often use the term "vacua" for the degenerate points, but at g>1 this is misleading. A lot of physics can hide at codimension-1 singularities of the CB. Can the author explain better why they expect that looking at 1-parameter families indeed fully captures the IR physics?

  2. It is not clear that the "point at infinity" F_\infinity is a well-defined concept in the full CB, as the degeneration pattern might be very different in different directions. Can the author elaborate on whether they expect the properties of section 5 to be generic (for any 1-dimensional slice) and why?

  3. Can the author explain how the result for 6d theory in section 5.3 is compatible with the rank-1 case, where the elliptic fiber of 6d N=(0,1) theories on the circle is trivial?

  4. Could the author explain how the proposed classification compares and is (or not) compatible with other classifications schemes in the recent literature. E.g. rank-2 works by Argyres-Martone, various works by Cecotti (including Caorsi-Cecotti), works on global forms of 4d N=2 SQFTs by Argyres-Martone-Ray and Closset-Magureanu, etc. Actively engaging with "competing" works would make the paper so much stronger.

Recommendation

Ask for minor revision

  • validity: good
  • significance: high
  • originality: high
  • clarity: low
  • formatting: acceptable
  • grammar: below threshold

---

## Editorial Decision

awaiting_resubmission